# *VTDexManip*: A Dataset and Benchmark for Visual-tactile Pretraining and Dexterous Manipulation with Reinforcement Learning

**Qingtao Liu, Yu Cui, Zhengnan Sun, Gaofeng Li, Jiming Chen, Qi Ye**[*]
Department of Control Science and Engineering
Zhejiang University
Hangzhou, Zhejiang, China
{l_qingtao, ccuiyu, zn.sun, gaofeng.li, cjm, qi.ye}@zju.edu.edu

## Abstract

Vision and touch are the most commonly used senses in human manipulation. While leveraging human manipulation videos for robotic task pretraining has shown promise in prior works, it is limited to image and language modalities and deployment to simple parallel grippers. In this paper, aiming to address the limitations, we collect a vision-tactile dataset by humans manipulating 10 daily tasks and 182 objects. In contrast with the existing datasets, our dataset is the first visual-tactile dataset for complex robotic manipulation skill learning. Also, we introduce a novel benchmark, featuring six complex dexterous manipulation tasks and a reinforcement learning-based vision-tactile skill learning framework. 18 non-pretraining and pretraining methods within the framework are designed and compared to investigate the effectiveness of different modalities and pertaining strategies. Key findings based on our benchmark results and analyses experiments include: 1) Despite the tactile modality used in our experiments being binary and sparse, including it directly in the policy training boosts the success rate by about 20% and joint pretraining it with vision gains a further 20%. 2) Joint pretraining visual-tactile modalities exhibits strong adaptability in unknown tasks and achieves robust performance among all tasks. 3) Using binary tactile signals with vision is robust to viewpoint setting, tactile noise, and the binarization threshold, which facilitates to the visual-tactile policy to be deployed in reality. The dataset and benchmark are available at `https://github.com/LQTS/VTDexManip`.

## 1 Introduction

Achieving human-like dexterous manipulation is one of the most challenging tasks in robotics and embodied artificial intelligence. When humans interact with the world, vision and touch are the common senses used. Touch offers physical information beyond vision and compensates for local perception when vision is obstructed Billard & Kragic (2019). Therefore, extensive studies Lee et al. (2020); Chen et al. (2022); Qi et al. (2023); Zhang & Demiris (2023); George et al. (2024); Calandra et al. (2018); Jin et al. (2024) attach tactile sensors to grippers and leverage both visual and tactile signals in a reinforcement learning framework for skill learning. However, the visual-tactile representation is trained jointly with the manipulation policy for a specific task in limited environments, struggling to generalize to other tasks and environments.

On the other hand, representation learning on large datasets has demonstrated generalization abilities in many domains Radford et al. (2021); Zitkovich et al. (2023); Driess et al. (2023); Achiam et al. (2023). To address the generalization issue in robotic manipulation, some works have collected extensive robot operation data to train large robotic models Brohan et al. (2022); Dasari et al. (2019); Bharadhwaj et al. (2023); Fang et al. (2023); Khazatsky et al. (2024). However, the data collection cost is high, and it is challenging to gather complex manipulation skills on a large scale, resulting in these datasets focusing on tasks with simple parallel grippers. In contrast, humans can perform

---

[*]Coresponing Author

complex dexterous manipulation tasks, such as in-hand rotation and it is easier to acquire large-scale human manipulation videos. Therefore, recent studies have explored the representations learned from large datasets of human manipulation for robotic tasks Radosavovic et al. (2023); Nair et al. (2022); Karamcheti et al. (2023); Ma et al. (2023b); Zhang et al. (2023); Ma et al. (2023a). Nevertheless, they primarily emphasize the visual and natural language modalities of human manipulation data, neglecting the potential benefits of tactile data.

To bridge the gap, we create a dataset of 2032 vision-tactile sequences covering 10 daily tasks and 182 objects to study the tactile modality and multi-modal pretraining in complex manipulations. Recently, attention has been paid to the fusion of vision and tactile modalities and several visual-tactile datasets are collected to perform tasks like material classification, tactile localization, object property prediction, garment classification Yang et al. (2022); Kerr et al. (2023); Dou et al. (2024); Yu et al. (2024b) as shown in Tab.1. For our manipulation data collection, humans need to wear tactile sensors to collect data for a long time and the sensors must be flexible and robust while these optical-based tactile sensors are typically bulky and can not be easily integrated into wearable gloves or attached to human hands at low cost. On the other hand, though these high-precision tactile signals help sophisticated control at the low level, the combination and the tempo of touch status of different hand parts may provide abundant information on *how* to manipulate in a higher planning level. As a compromise, we adopt a solution of low-cost piezoresistive pressure sensors attached to a glove as shown in Fig.1(a). Also, in contrast with the existing visual-tactile datasets, our dataset is the first visual-tactile dataset with multi-fingered hand for complex robotic manipulation skill learning.

Table 1: Comparison of visual-tactile datasets.

| Dataset | #Frames/Objs/Seqs/Tasks | Vision | Tactile | Application |
|---|---|---|---|---|
| Touch and go Yang et al. (2022) | 13.9K/3971/-/- | RGB | GelSight | Tactile-driven image stylization, multimodal video prediction |
| SSVTP Kerr et al. (2023) | 4500/10/-/- | RGB | DIGIT | Anomaly detection, edge following, tactile localization and classification |
| TaRF Dou et al. (2024) | 19.3K/-/-/- | RGB | DIGIT | Tactile localization, material classification |
| PHYSICLEAR Yu et al. (2024b) | 45.8k/76/408/- | RGB-D | GelSight | Physical property prediction, scenario reasoning |
| Ours | **565k**/182/**2032/10** | RGB | Pressure sensor | Complex multi-fingered manipulation tasks |

To evaluate the role of human vision-tactile information in robotic manipulation, we introduce a novel benchmark comprising a vision-tactile dexterous manipulation platform based on Isaac Gym Makoviychuk et al. (2021) and a reinforcement learning-based skill learning framework, with the hope of providing insights for future research relating to dexterous multi-fingered robotic manipulation, human priors for robotic manipulation, multi-modal learning, tactile sensor design for multi-fingered hands, etc. . Specifically, six complex manipulation tasks are built up in the simulation that require sophisticated coordination of joint movements: Bottlecap Turning, Faucet Screwing, Lever Sliding, Table Reorientation, In-hand Reorientation and Bimanual Hand-over. Though there exists work on some similar tasks like Reorientation Chen et al. (2023), they only focus on one task. Our multi-fingered manipulation platform is the first one consisting of very different types of complex skills, which facilitates the study of generalizable human-like complex manipulation skill learning.

For the benchmark, a benchmark method fusing vision-tactile via joint pretraining is first constructed to learn manipulation representations from human vision-tactile data inspired by MAE He et al. (2022); Radosavovic et al. (2023); Liu et al. (2024). Further, five popular vision pretraining methods Radosavovic et al. (2023); Nair et al. (2022); Radford et al. (2021); He et al. (2016); Karamcheti et al. (2023) used in computer vision and robotics and their combination with the tactile modality are benchmarked. In addition to the methods, experiments and analyses regarding the viewpoint setting for visual-tactile fusion, the tactile noise, and thresholds to binarize the tactile signals in pretraining and RL are conducted to provide guidance for deploying the tactile modality to real-world experiment setting. Our main contributions are summarized as follows:

- We collect a human visual-tactile manipulation dataset consisting of 565k frames, covering 10 daily tasks and 182 objects for multi-fingered robotic hand manipulation.

- We propose a vision-tactile benchmark for dexterous manipulation, which includes a manipulation simulation platform with six multi-fingered manipulation tasks and a manipulation skill learning framework based on pretraining and reinforcement learning.

- We benchmark more than 18 non-pretraining and pretraining methods to investigate the effectiveness of different modalities and pretraining strategies for dexterous manipulation.
- Extensive experiments and analyses regarding viewpoint setting, the noise and binary thresholds of the tactile signals are conducted to provide guidance for the deployment of the tactile modality.

## 2 RELATED WORKS

**Visual-Tactile datasets and benchmarks.** In recent years, the integration of visual and tactile sensing has gained increasing attention in robotics, particularly for tasks that involve intricate surface interactions. Optical tactile sensors Lambeta et al. (2020); Yuan et al. (2017), in particular, have enabled the collection of rich visual-tactile data that captures detailed texture information from the contact area. With these sensors, prior works collected datasets that have been primarily used for tasks such as texture recognition, classification, localization, and detection Yang et al. (2022); Kerr et al. (2023); Dou et al. (2024); Yu et al. (2024b). While visual-tactile information plays a crucial role in dexterous manipulation Qi et al. (2023); Guzey et al. (2023), there is currently no dataset available specifically tailored for such tasks. The use of human tactile information for robot manipulation has gained increasing attention. MimicTouch Yu et al. (2024a) leverages human tactile demonstrations for gripper-based tasks like grasping and insertion. Inspired by Liu et al. (2024), our work fills this gap by providing a visual-tactile dataset for learning diverse complex dexterous manipulation skills with multi-fingered hands.

**Pretraining for robot manipulation with out-of-domain dataset.** Due to the challenges of collecting robotic data and generalizing in-domain data, some previous works Radosavovic et al. (2023); Nair et al. (2022); Karamcheti et al. (2023); Ma et al. (2023b); Zhang et al. (2023); Ma et al. (2023a) have focused on leveraging out-of-domain datasets for pretraining. These works have shown that representation models pretrained on human data can significantly aid in learning different robotic tasks. These works are mainly divided into two categories: MVP Radosavovic et al. (2023), R3M Nair et al. (2022), Voltron Karamcheti et al. (2023), and SGR Zhang et al. (2023) pretrain models to extract representations from inputs, while VIP Ma et al. (2023b) and LIV Ma et al. (2023a) pretrain representation and reward function models. All these works utilize human manipulation datasets Grauman et al. (2022); Damen et al. (2018); Goyal et al. (2017), focusing on visual, language, or point cloud modalities. Tactile sensing, despite its crucial role in both human and robotic manipulation, has not been explored extensively due to the absence of human tactile data in existing datasets.

**Visual-tactile manipulation.** Many studies have attached tactile sensors to parallel grippers and combined the tactile signals with vision to realize manipulation tasks like cable plugging George et al. (2024), dressing Zhang & Demiris (2023), efficient grasp adjustment Calandra et al. (2018), and insertion Jin et al. (2024); Sferrazza et al. (2023). For multi-finger manipulation tasks, Guzeyet al. Guzey et al. (2023) places uSkin on the Allegro Hand to learn six dexterous manipulation skills. RotateIt Qi et al. (2023) equip robotic fingertips with four omnidirectional vision-based touch sensors to learn the in-hand rotation skill. These studies focus on online learning of visual and tactile modalities, achieving promising results for specific tasks. However, the learned representation models are not transferable to other tasks, requiring task-specific data and representation learning for each case, which is inefficient for multi-skill learning. Instead, we mainly study and analyze the efficacy of visual and tactile pretraining for skill learning on various complex manipulation tasks by human manipulation data collection and pretraining benchmarking similar to Liu et al. (2024).

## 3 VISUAL-TACTILE MANIPULATION DATASET

**Data Collection Procedure.** Following Liu et al. (2024), we set up our collection system as shown in Fig.1(a). We calibrate each tactile unit on the glove using an F/T sensor to ensure that each tactile unit can be activated by the same amount of force. Due to the simple principle and design, the glove can produced in a more compact form in industry, suitable for large-scale data collection. We utilize the glove and a Hololens2 [1] to synchronously collect visual and tactile data pairs of human manipulation from an ego-centric view. See A.1 for more about the glove.

**Dataset Statistics and Analysis.** The dataset is collected by 5 subjects and includes 2,032 manipulation sequences, 10 daily tasks, 182 objects, which ends up with 565k visual-tactile pairs of frame.

---

[1] https://www.microsoft.com/en-us/hololens

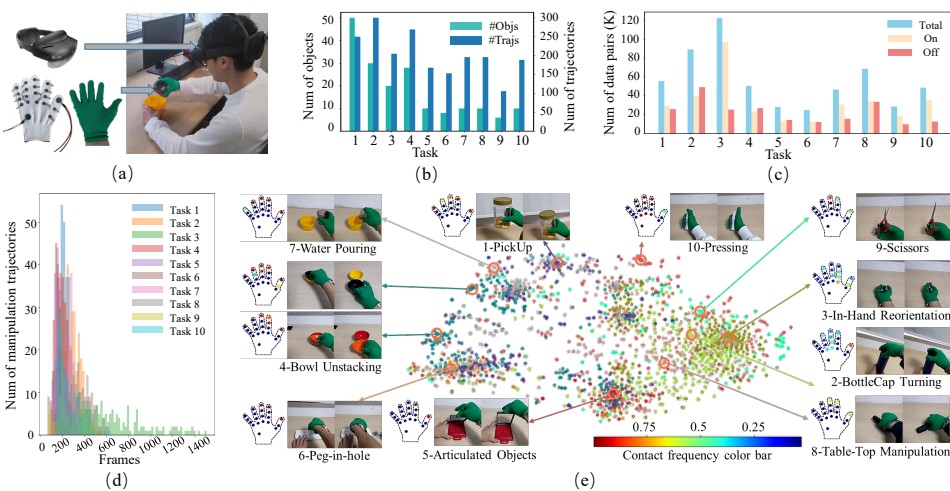

Figure 1: Visualization of our dataset. (a) Our collection system. (b) The number of trajectories and objects. (c): The number of total frames (On: frames w/ contact; Off: frames w/o contact ). (d) The distribution of the number of frames. (e) t-SNE of the tactile signals.

We show the number of trajectories and objects for each task in Fig.1(b), the number of total frames for each task in Fig.1(c), and the distribution of the number of frames in each task trajectory in Fig.1(d). The tactile signals carry pressure values during manipulation, but to reduce the real-to-sim and sim-to-real gap, only binary signals acquired by thresholding pressure values are used in the following sections. See A.2 for more details.

To further investigate the characteristics of the tactile signals for different manipulation tasks, we visualize the tactile signals by t-SNE Van der Maaten & Hinton (2008) as shown in Fig.1(e): for each sequence, we calculate the contact frequency of a tactile sensor and the frequency of all tactile sensors is concatenated as the representation of the sequence for visualization (See A.3 for more details about t-SNE). In the figure, there exist some clusters while a cluster does not correspond to only a task: 1) the clusters may consist of data points from different tasks, e.g. data points for In-hand Reorientation and BottleCap Turning form a relatively compact cluster; 2) data points for a task may span many clusters, e.g. data points for Bowl Unstacking scatter across different clusters.

The complex distribution however may inspire research in many aspects. The tactile clusters exhibited in our dataset may facilitate research in the dynamic manipulation taxonomy to understand human behaviors using tactile signals, not just the hand poses used Feix et al. (2016) for grasp taxonomy study. Second, the hand movement is continuous while the touch is not: the subtle changes of hands in the moment of contact are hard to be reflected in the images and spacial displacement while the discrete nature of the tactile signal can capture distinct information of finger gait and the synergy during hand-object interaction like opening and closing scissors represented in Fig.1(e), which can serve as a prior for complex manipulation skill learning. Also, the visual modality between different tasks often has a large difference, but the similarity presented in touch (e.g.Bowl Unstacking and Peg-in-hole as depicted in Fig.1(e).) can provide a basis for the generalization of task-level strategies in downstream tasks.

## 4 VISUAL-TACTILE DEXTEROUS MANIPULATION BENCHMARK

### 4.1 VISUAL-TACTILE MANIPULATION PLATFORM AND TASKS

In this section, we introduce our visual-tactile dexterous manipulation platform based on Isaac Gym environment Makoviychuk et al. (2021), which encompasses six complex tasks with a Shadow Hand Sharma et al. (2014). See B.1 for more details.

- **BottleCap Turning.** Agents are required to rotate the bottle cap counterclockwise by one full circle. 10 objects and 5 unseen objects from Wang et al. (2023) are prepared for skill learning and for evaluation. The task is seen in human hand manipulation data.

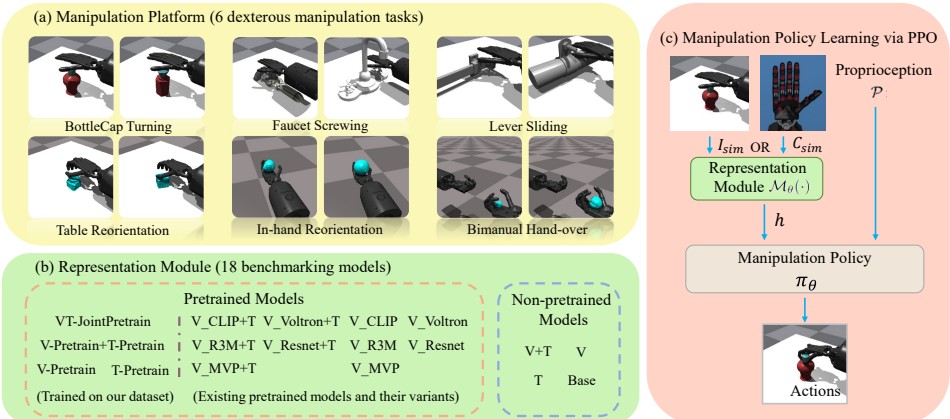

Figure 2: Overview of our benchmark. (a) shows the six tasks of our manipulation platform (Sec.4.1); (b) lists the 18 pretrained and non-pretrained models in our benchmark (Sec.4.2); (c) is the policy learning framework combines proprioceptive inputs and perception representations to guide actions within an MDP, with skills learned via PPO (Sec.4.3).

- **Faucet Screwing.** Similar to the BottleCap Turing, agents are required to rotate the tap handle one full circle clockwise. The task is unseen in human hand manipulation data. Five faucet models with a rotational axis perpendicular to the ground from the SAPIEN dataset Xiang et al. (2020) for manipulation policy learning and generate 10 test objects by scaling the taps using two different scaling factors for evaluation.

- **Lever Sliding.** Agents focus on actions to separate a long hole and an axe, which requires the fingers to press or pinch the object and use wrist motion to slide the axis out. We create 5 CAD models based on everyday objects and generate URDF (Unified Robot Description Format) files for them. We obtain 10 test objects by scaling the CAD models to two different sizes.

- **Table Reorientation.** This task enables the dexterous hand to learn how to rotate an object on the table without toppling it. Unlike the previous rotation task, the object in this task is not fixed. We use 10 objects for training and 5 objects for testing from the YCB dataset Calli et al. (2015).

- **In-hand Reorientation.** Agents are designed to rotate an object within the fingers while keeping the palm facing upward, aiming to rotate the object anticlockwise over half a circle without deviating beyond a specified threshold. 10 objects from the YCB dataset Calli et al. (2015) are used for training, and 5 objects are used for evaluation.

- **Bimanual Hand-over.** This is a bimanual manipulation task in which one hand throws an object while the other catches it, ensuring the object does not drop. This task primarily trains the dexterous hand in bimanual coordination. We use 5 objects for training and 3 objects for testing from the YCB dataset Calli et al. (2015).

**Acquisition of Visual and Tactile Sensing.** Isaac Gym Makoviychuk et al. (2021) provides users with a rich set of APIs for visual and force sensors. We position ego-centric cameras to capture RGB images of dexterous manipulation in the simulation as shown in the manipulation platform of Fig.2. Similar to the collection of human data, we arrange 20 force sensors on the dexterous hand corresponding to the tactile glove as shown in the right column of Fig.2.

## 4.2 VISUAL-TACTILE PRETRAINED MODELS

We aim to study whether human visual-tactile priors can enhance robotic dexterous manipulation skill learning. We benchmark various pretrained and non-pretrained methods for visual and/or tactile representation. More details for these methods in B.2 of the supplementary material.

**Visual-tactile Fusion by Joint Pretraining.** Inspired by the masking reconstruction pretraining He et al. (2022); Radosavovic et al. (2023); Liu et al. (2024), we construct a benchmark method **VT-JointPretrain** to fuse the visual and tactile modalities, which consists of a fusion encoder and a reconstruction decoder. The fusion encoder $E_\theta$ integrates image-tactile pairs $(V, C)$. The RGB image $V$ is divided into patches and transformed into embeddings $\bar{v}$ and tactile data $C$ is sliced into

patches and converted into embeddings $\bar{c}$. We randomly mask patches in a certain proportion and use a transformer encoder to fuse the visible patch embeddings along with a learnable CLS token, producing a fused representation $\{h_v, h_c, h_{CLS}\}$. The CLS token aggregates the latent features of the visual-tactile pair and serves as the input to the downstream task network. For the reconstruction decoder, $\{h_v, h_c\}$ and the masked tokens are input into the transformer decoder. We then use $R_{\theta_v}$ to project $\hat{v}$ to the reconstructed image $\hat{V}$ and $R_{\theta_c}$ to map each reconstructed tactile patch $\hat{c}_i$ to their reconstructed unit $\hat{C}_i$. The network is trained with the discrepancy of the reconstructed images and tactile signals with the original ones. After convergence, the encoder $E_\theta$ is leveraged for downstream tasks and the token $h_{CLS}$ is input to a policy network. Based on the method, we create two single-modality pretrained structures by removing either modality, resulting in **V-Pretrain** and **T-Pretrain**. Both methods are trained using our dataset. Also, we construct another baseline **V-Pretrain+T-Pretrain** by concatenating the features of **V-Pretrain** and **T-Pretrain**.

**Other Pretrained Models.** We collect 5 pretrained models: CLIP Radford et al. (2021), R3M Nair et al. (2022), MVPRadosavovic et al. (2023), VoltronKaramcheti et al. (2023), ResNet18He et al. (2016). We employ their open-source visual encoders to the RL framework, named as **V_CLIP**, **V_R3M**, **V_MVP**, **V_Voltron** and **V_Resnet**. We further extend the five methods by adding tactile modality feature extraction with an MLP, concatenating the visual and tactile features to form the visual-tactile representation. We name their variants as **V_CLIP+T**, **V_R3M+T**, **V_MVP+T**, **V_Voltron+T** and **V_Resnet+T**.

**Non-pretrained Methods.** We use the network structure of ResNet18 to extract image features and an MLP to extract tactile features, forming three non-pretrained baseline models. **T** uses only tactile input, **V** uses only image input, and **V+T** uses both image and tactile information, concatenating their features to form the visual-tactile representation. Additionally, we prepare a **Base** baseline model that uses only the proprioceptive information of the dexterous hand as input.

## 4.3 Manipulation Policy Learning

We model the dexterous manipulation task as a Markov Decision Process (MDP), defined by a tuple: $(\mathcal{S}, \mathcal{A}, \mathcal{T}, \mathcal{R}, \gamma)$. We use the PPO algorithm Schulman et al. (2017) to make the agent learn manipulation skills. The architecture is shown in the right column of Fig.2. We define the state as $S = \{h \leftarrow \mathcal{M}_\theta(\cdot), \mathcal{P}\}$ and the action and policy as $a = \pi_\theta(S)$. $\mathcal{M}_\theta(\cdot)$ is a representation module, which can be the pretrained encoders above or non-pretrained ones. For the pretrained encoders, $\mathcal{M}_\theta$ is frozen during policy training. $\mathcal{M}_\theta$ takes the ego-centric RGB image $V_{sim}$ or the binarized tactile signals $C_{sim}$ as input and generates the perceptual representation $h$. $V_{sim}$ is captured by the ego-centric camera in the simulator and $C_{sim}$ is the binarized result of the signal obtained from the force sensors, with a tactile threshold set to 0.01 N across all tasks. $\mathcal{P}$ represents the proprioceptive information of the dexterous hand, which includes the joint angles and joint velocities of the hand. See B.3 and B.4 for more details about RL training for each task.

## 5 Benchmarking Study

Table 2: Success rate (%) for our method and baselines with different modalities in all tasks.

| Tasks | Split | Base | T-Pretrain | V-Pretrain | **VT-JointPretrain** |
|---|---|---|---|---|---|
| BottleCap Turning | Seen | 55.9± 5.6 | 75.4± 2.9 | 70.8± 7.2 | **83.7± 0.9** |
| | Unseen | 36.8± 9.4 | 68.6± 5.6 | 58.5±14.2 | **81.3± 0.5** |
| Faucet Screwing | Seen | 49.0±12.0 | 60.0±12.3 | 57.9± 7.0 | **80.1± 1.8** |
| | Unseen | 43.9±10.5 | 51.9±12.1 | 51.8± 6.5 | **73.6± 2.1** |
| Lever Sliding | Seen | 5.8 ± 4.4 | 53.1±23.1 | 27.9±14.9 | **89.3± 3.6** |
| | Unseen | 2.2 ± 1.9 | 48.3±20.7 | 20.5±10.9 | **79.6± 6.1** |
| Table Reorientation | Seen | 51.8± 6.3 | 68.8± 1.8 | 74.2± 9.4 | **85.0± 1.4** |
| | Unseen | 46.7± 7.3 | 69.8± 2.3 | 69.2±10.0 | **84.6± 1.1** |
| In-hand Reorientation | Seen | 38.1± 2.4 | 42.1± 2.7 | 55.7± 1.5 | **62.2± 5.0** |
| | Unseen | 33.7± 1.6 | 35.8± 2.6 | 53.5± 1.7 | **55.1± 2.7** |
| Bimanual Hand-over | Seen | 8.0 ± 4.4 | 35.0±10.2 | 37.7±10.9 | **45.5± 1.5** |
| | Unseen | 3.3 ± 1.4 | 20.7± 6.0 | 23.1± 7.0 | **26.6± 1.9** |
| Task Mean | Seen | 34.8± 5.8 | 55.7± 8.8 | 54.0± 8.5 | **72.2± 2.4** |
| | Unseen | 27.8± 5.0 | 49.2± 9.4 | 46.1± 8.1 | **66.8± 2.7** |

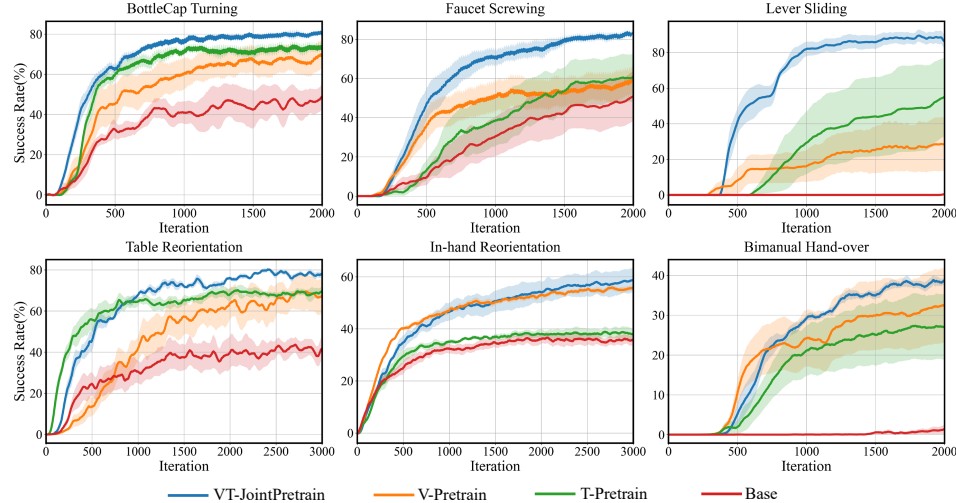

Figure 3: The training process of different modalities in all tasks.

In this section, we first investigate the effect of the tactile information and compare different pretrained and non-pretrained methods. Then, we study the robustness of methods under different viewpoints, analyze the impact of the tactile threshold of the pretraining and RL, and investigate the impact of the tactile noise on the policy learning and deployment, which are important aspects for exploiting the tactile modality. In all experiments, the metric for all tasks is ***Success Rate (%)***. If the dexterous hand achieves the goals in one episode, it is considered successful. All experiments use 4 random seeds, and the results are tested 100 times for seen and unseen object. All baselines are described in Sec.4.2.

## 5.1 EFFECTIVENESS OF TACTILE INFORMATION.

We conduct five groups of experiments in all six tasks. The training process is shown in Fig.3 and the evaluation results are listed in Tab.2. With the tactile informaiton, **T-Pretrain** demonstrates significant improvements over the **Base** method. Furthermore, compared to the **V-Pretrain** methods that utilize only vision, the **VT-JointPretrain** methods that incorporate extra tactile modality features show considerably better performance. The results show that incorporating tactile information significantly improves the learning of manipulation skills, even across different types of tasks.

## 5.2 BENCHMARKING PRETRAINED AND NON-PRETRAINED METHODS.

Table 3: Success rate (%) for benchmarking different methods.

| Method | Modality | Pretrain | Joint pretrain | Seen | Unseen |
|---|---|---|---|---|---|
| T | t | ✗ | - | 50.8± 2.5 | 47.0± 2.1 |
| V | v | ✗ | - | 24.0± 3.0 | 22.2± 2.9 |
| V+T | v+t | ✗ | - | 23.6± 2.6 | 19.3± 2.9 |
| V-MVP | v | ✓ | - | 35.2± 2.7 | 29.4± 2.4 |
| V-Voltron | v | ✓ | - | 40.0± 1.9 | 31.7± 1.5 |
| V-R3M | v | ✓ | - | 37.0± 0.7 | 26.2± 2.1 |
| V-CLIP | v | ✓ | - | 61.3± 1.5 | 49.4± 1.8 |
| V-ResNet | v | ✓ | - | 54.1± 0.5 | 46.8± 0.6 |
| V-MVP+T | v+t | ✓ | ✗ | 38.5± 2.5 | 35.3± 2.3 |
| V-Voltron+T | v+t | ✓ | ✗ | 39.8± 2.1 | 34.7± 2.0 |
| V-R3M+T | v+t | ✓ | ✗ | 38.9± 2.1 | 31.0± 1.5 |
| V-CLIP+T | v+t | ✓ | ✗ | 65.4± 1.7 | 55.9± 1.7 |
| V-ResNet+T | v+t | ✓ | ✗ | 55.4± 1.9 | 44.1± 1.8 |
| V-Pretrain+T-Pretrain | v+t | ✓ | ✗ | 62.6± 6.3 | 53.3± 7.3 |
| VT-JointPretrain | v+t | ✓ | ✓ | **74.3± 0.6** | **65.7± 0.7** |

We set up three groups of methods for benchmarking in all six tasks, shown in Tab.3. The first is non-pretrained models, taking only tactile (**T**), only vision (**V**), or vision and tactile (**V+T**) as input. The second group is the vision-pretrained models used in computer vision and robotics, which are MVP (**V_MVP**), Voltron(**V_Voltron**), R3M(**V_R3M**), CLIP(**V_CLIP**), and ResNet(**V_ResNet**). Their features are concatenated with tactile features extracted by the method **T** to generate visual-tactile pretrained methods (**V_MVP+T, V_Voltron+T, V_R3M+T, V_CLIP+T, V_ResNet+T**). The

last group is the visual-tactile concatenation method (**V-Pretrein+T-Pretrain**) and fusion method (**VT-JointPretrain**) pretrained with our dataset.

It can be observed that, regardless of whether pretrained or non-pretrained methods are used, most approaches relying solely on visual inputs show improved performance when augmented with tactile information. However, directly concatenating tactile features is less effective than joint training with **VT-JointPretrain**. Similarly, combining separately pretrained visual and tactile models (**V-Pretrain+T-Pretrain**) also yields suboptimal results. In contrast, **VT-JointPretrain** effectively fuses visual and tactile signals by masking input modalities and recovering the original signals, leading to significant performance gains. Additionally, joint pretraining substantially reduces result variance across different tasks and training random seeds, indicating improved consistency and robustness.

## 5.3 ANALYSIS OF VISUAL AND TACTILE MODALITIES

In this section, we further study the characteristics of the binary tactile signals of our dataset using the visual-tactile fusion method **VT-JointPretrain** in the BottleCap Turning task.

**1). Viewpoint adaptability for RL.** We study the robustness to changes in perspective by further testing the camera mounted on the robot arm (**\*-arm**) and a third-person view (**\*-3rd**). **\*-ego** refers to the perspective used in previous experiments. Tab.4 shows the average success rates for the success rates under each perspective for seen and unseen objects. Adding tactile information helps to improve the average success rate and also reduces the variations due to perspectives.

**2). Robustness to tactile thresholds for RL.** We study the robustness to different tactile thresholds. We set tactile thresholds to 0.5N (**\*-50**) and 1N (**\*-100**). 0.01N(**\*-1**) is the previous threshold setting. Similarly, Tab.4 shows the average success rates for the success rates of different thresholds. The performance of both tactile-only and visual-tactile methods declines as the threshold increases.

Table 4: Success rate (%) of different visual and tactile modalities.

| Method | Vision viewpoints | Tactile thresholds | Seen | Unseen |
|---|---|---|---|---|
| V-ego (V-Pretrain) | ego-centric | - | 70.8± 7.2 | 58.5±14.2 |
| V-arm | on the arm | - | 58.2±16.9 | 58.5±17.0 |
| V-3rd | third view | - | 46.2±17.6 | 37.4±18.8 |
| VT-arm | on the arm | 0.01N | 78.0± 4.9 | 73.3± 7.0 |
| VT-3rd | third view | 0.01N | 82.4± 2.3 | 79.4± 4.2 |
| T-1 (T-Pretrain) | - | 0.01N | 75.4± 2.9 | 68.6± 5.6 |
| T-50 | - | 0.5N | 60.8± 9.1 | 48.8±14.4 |
| T-100 | - | 1.0N | 64.3± 5.9 | 46.6± 9.7 |
| VT-50 | ego-centric | 0.5N | 82.9± 1.2 | 80.6± 0.2 |
| VT-100 | ego-centric | 1.0N | 74.4± 4.8 | 65.0± 8.8 |
| VT-JointPretrain | ego-centric | 0.01N | **83.7± 0.9** | **81.3± 0.5** |

**3). Impact of tactile threshold on pretraining.** In addition to the threshold of 0.2V discussed in the paper, we conduct two additional experiment settings: pretrain **VT-JointPretrain** using threshold voltages of 0.55V and 0.75V for real tactile signals to study the sensitivity of the joint pretraining method to the tactile threshold. According to the calibration data from the tactile glove, described in A.1 of the supplementary materials, these thresholds correspond to forces of 0.05 N, 0.5 N, and 1 N, respectively. We use these pretrained models to conduct RL training with three different simulation force thresholds (0.01 N, 0.5 N, and 1.0 N). Tab.5 presents the results of these experimentsk. The success rate of the policy only declines significantly when the threshold difference between real data pretraining and simulation RL training is as large as 20-fold (0.05 N vs. 1.0 N). These results demonstrate that using binary tactile signal is robust to the thresholds used for real data, and mismatches between the thresholds in real data and simulation.

**4). Impact of tactile noise for policy learning and deployment.** To assess the impact of tactile noise on downstream tasks, we add Gaussian noise with standard deviations of 0.01N, 0.1N, and 1.0N to tactile signals, keeping the binarization threshold at 0.01N. We train the policy without noise and test it under tactile noise (**v1**) and hysteresis noise (**v2**) with thresholds ranging from 0.01N to 0.5N. Additionally, we train a policy with tactile signals augmented by Gaussian noise ($\sigma = 0.1$) and test it with varying noise levels (**v3**). The results (Tab.6) show that models trained without noise are sensitive to tactile noise during testing, while applying a hysteresis threshold significantly reduces this effect, suggesting that pretrained models using binary tactile signals can be effectively deployed

Table 5: Tactile threshold setting for pertaining

|  | Force threshold (N) for RL training | | | | | |
|  | 0.01N | | 0.5N | | 1.0N | |
|  | Seen | Unseen | Seen | Unseen | Seen | Unseen |
| 0.2V (0.05N) | 83.7± 0.9 | 81.3± 0.5 | 82.9± 1.2 | 80.6± 0.2 | 74.4± 4.8 | 65.0± 8.8 |
| 0.55V (0.5N) | 80.5± 6.7 | 77.8± 6.3 | 85.3± 5.1 | 82.3± 2.1 | 82.5± 4.3 | 81.6± 4.2 |
| 0.75V (1.0N) | 81.6± 3.5 | 79.3± 5.1 | 86.4± 3.8 | 84.2± 3.0 | 80.8± 4.4 | 77.8± 7.2 |

Table 6: Tactile noise setting for RL

|  | $\sigma$=0.01N | | $\sigma$=0.1N | | $\sigma$=1N | |
|  | Seen | Unseen | Seen | Unseen | Seen | Unseen |
| v1 | **60.3± 7.8** | **62.7± 6.0** | 37.0± 10.9 | 33.7± 6.4 | 34.4± 10.2 | 33.5± 8.0 |
| v2 | **83.8± 1.4** | **81.2± 0.8** | 79.1± 1.4 | 79.9± 2.2 | 41.2± 10.1 | 40.7± 4.8 |
| v3 | 84.6± 4.6 | 81.7± 9.6 | **87.1± 4.2** | **84.6± 7.8** | 86.8± 5.5 | 83.8± 8.1 |

Table 7: Success rate (%) for deploying single modality after joint pretraining to RL

| Method | V-Pretrain | VT-JointPretrain-MaskT | T-Pretrain | VT-JointPretrain-MaskV | VT-JointPretrain |
|---|---|---|---|---|---|
| Seen | 70.8± 7.2 | 73.3± 2.9 | 75.4± 2.9 | 72.6± 2.7 | 83.7± 0.9 |
| Unseen | 58.5±14.2 | 65.7± 5.0 | 68.6± 5.6 | 66.1± 7.0 | 81.3± 0.5 |

in real-world scenarios by adjusting the binarization schedule. Furthermore, introducing noise during RL training not only improves performance but also enhances robustness to noise. Notice that the average force is about 10N to 15N for different tasks manipulation. The setting of thresholds and stand deviations from 0.01N to 1N spans a large range.

**5). Deploy single modality to RL after joint pretraining.** In this part, we investigate whether only using one modality after joint pretraining can benefit from the modality fusion. Therefore, we mask the input visual and tactile modality of the joint pretrained model separately, denoted as **VT-JointPretrain-MaskV** and **VT-JointPretrain-MaskT**. In Tab.7, comparing **VT-JointPretrain-MaskT** with **V-Pretrain**, the fusion algorithm demonstrates superior performance in downstream tasks when only visual input is provided, indicating the tactile information contributes to the visual feature extraction during joint pretraining. However, similar distillation is not observed when masking visual modality. **VT-JointPretrain-MaskV** even sees a little drop in the success rate.

## 5.4 REAL-WORLD EXPERIMENTS

The real-world experiment employs a Shadow Hand Sharma et al. (2014) and an Azure Kinect camera[2]. To collect tactile data, 20 piezoresistive sensors are attached across the Shadow Hand. The hardware and control algorithm are integrated via ROS, enabling the collection of environmental states and observations. To address the Sim2Real gap, a teacher policy is first trained using domain randomization Tobin et al. (2017) on joint angles, velocities, and actions, with Gaussian noise (standard deviation 0.1) added to tactile forces before binarization during RL training. This policy is then distilled into a student policy using Dagger Ross et al. (2011), incorporating augmented visual inputs. The tactile sensors are evaluated on the Shadow Hand, with 0.2V set as the binarization threshold. Further details and examples are provided in C of the supplementary material.

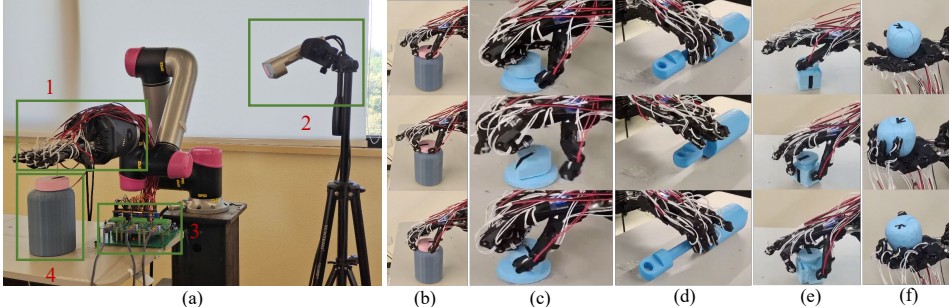

Figure 4: Real-world experiments. a) Hardware: 1-Shadow Hand, 2-Azure Kinect camera, 3-Tactile collection board. 4-Bottle. Deployment to real world: (b-f) BottleCap Turning, Faucet Screwing, Lever Sliding, Table Reorientation, In-hand Reorientation.

## 5.5 DISCUSSION, LIMITATION AND FUTURE WORK

**Downstream Task Adaptation.** Among the six tasks, only the BottleCap Turing task is in the dataset and the others are unseen tasks. The hand over task is distinct with two hands. Despite the difference, the model pre-trained with our visual-tactile data **VT-JointPretrain** performs exceptionally well on all tasks. This indicates that the visual-tactile pretraining approach not only improves performance on known tasks but also exhibits adaptability in unknown tasks, particularly in complex manipulation requiring tactile feedback. As shown in Tab.2, for the first four tasks where visual information of the

---

[2] https://azure.microsoft.com/en-us/products/kinect-dk

objects is prone to blocked by the robotic hands, tactile information significantly enhances model performance. However, in the last two tasks, where visual information is relatively complete, the advantages of visual input become more pronounced. The results demonstrates the complementary nature of visual and tactile information and the joint pretraining via MAE exploits the nature.

**Tactile Modality.** In the tactile research community, various tactile sensors have been developed and many of them can capture very high dimensional and dense information like images, e.g. GelSight Yuan et al. (2017). Pretraining these dense tactile patches and aligning them with images has been demonstrated effective for downstream tasks like physical reasoning Yu et al. (2024b). However, they have not been evaluated for dexterous manipulation with complex dynamics and coordination between fingers. Though not able to capture highly precise force information, tactile signals carry important information about the coordination between fingers during complex interactions with objects. We attribute the effectiveness of tactile modality demonstrated in the results to the embedded coordination information and also complementing images containing occlusions.

**Tactile Pretraining.** In pretraining, as discussed in **5)** of Sec.5.3, vision information fails to transfer effectively to the tactile modality after joint learning. Unlike the continuity of images and poses, tactile signals are inherently discrete, reflecting either contact or no contact. Effectively leveraging this discrete nature remains an open challenge. Our initial exploration focuses on binary tactile signals; future work must address signals with pressure values, shear forces, sensor sensitivity variations, and the simulation-to-reality gap, aiming for policies that generalize across tasks. Fully exploiting tactile signals also requires high-speed, high-fidelity simulations of complex object-sensor interactions—a major challenge for the field. Additionally, for analysis, the visual-tactile pretraining method **VT-JointPretrain** is trained from scratch on our dataset. Integrating large vision-language models with tactile data presents a promising avenue for enhancing performance.

**Cross-Modal Regularization for Dexterous Manipulation.** Integrating high-precision tactile modalities into robotic systems often increases cost and complexity. However, as shown in Tab. 7, our visual-tactile fusion approach demonstrates that even sparse tactile data can enhance visual representations by serving as a regularization mechanism. This underscores the potential of tactile signals to boost efficiency and robustness in multimodal learning. Future research could explore distilling tactile into visual modalities, achieving similar performance with reduced reliance on high-precision tactile hardware—lowering deployment costs without compromising accuracy.

**Adaptation to Various Tactile Sensors.** Learning a manipulation policy with different tactile sensors is a major challenge in robotics due to the diverse physical principles involved, such as capacitive, piezoresistive, or optical sensing. Though the pretraining methods evaluated in the work cannot deal with the original tactile signals, in principle they can work with the binarized tactile signals if a proper preprocessing thresholding step is applied to the original ones. With the thresholding, the differences of tactile sensors become a black box to these pretraining methods, and working with binarized signals mitigates issues like noise, scaling, and sim2real transfer.

**Robustness to Tactile Noise.** Tactile sensors can produce varying levels of noise when interacting with objects. MAE-based pretraining methods adopted in this work can mitigate this issue by masking the tactile signals during pretraining. The models are forced to learn how to infer missing information, thereby enhancing its robustness to noise and data loss. The denoising effect of MAE is also verified in Yuda Zou (2024), where MAE is used to denoising the key-point labeling noise.

# 6 CONCLUSION

In this study, we present the first visual-tactile dataset for complex manipulation skill learning and introduce a benchmark with six challenging tasks and a reinforcement learning-based framework. We evaluate 18 pretraining and non-pretraining methods to explore the effectiveness of different modalities and pretraining strategies for dexterous manipulation, conducting extensive experiments on various viewpoints, tactile thresholds, and noise levels. The results reveal key insights: 1) Incorporating binary, sparse tactile information significantly enhances complex dexterous skill learning, especially when combined with vision pretraining. 2) Joint visual-tactile pretraining shows strong adaptability to unseen tasks and achieves robust performance across all tasks. 3) Binary tactile signals exhibit high robustness to threshold variations and tactile noise, helping to mitigate the sim-to-real gap. Our work provides the support for advancing dexterous manipulation through visual and tactile modalities. We will release the benchmark and hope it can help find better solutions.

## 7 ACKNOWLEDGMENT

This work was supported by the National Natural Science Foundation of China (Grant No. 62088101, 62233013) and the Key Research and Development Program of Zhejiang Province (Grant No. 2025C01072). We sincerely appreciate Peisen Xu and Zhou Lu for their assistance in data collection, as well as Siyun Wang and Jiaying Chen for their support in conducting the physical experiments.

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

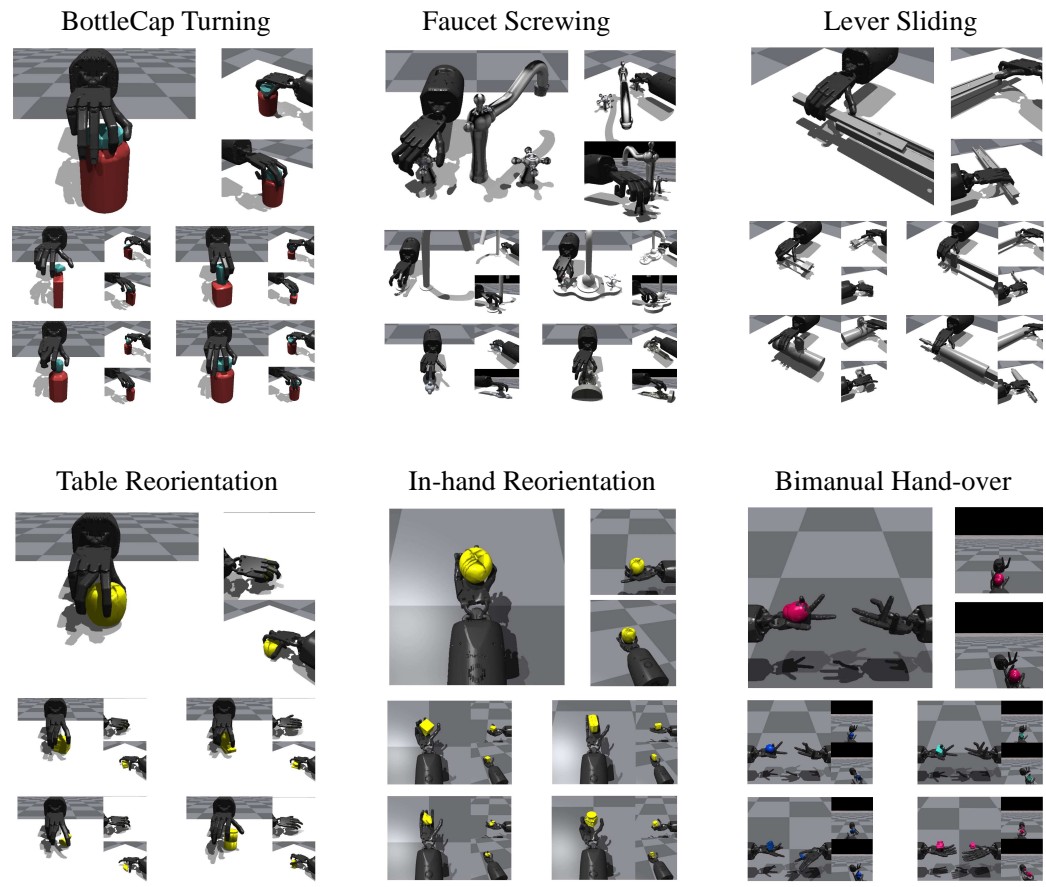

Figure 5: Visualizaiton of manipulation policies on different tasks and objects in the simulation

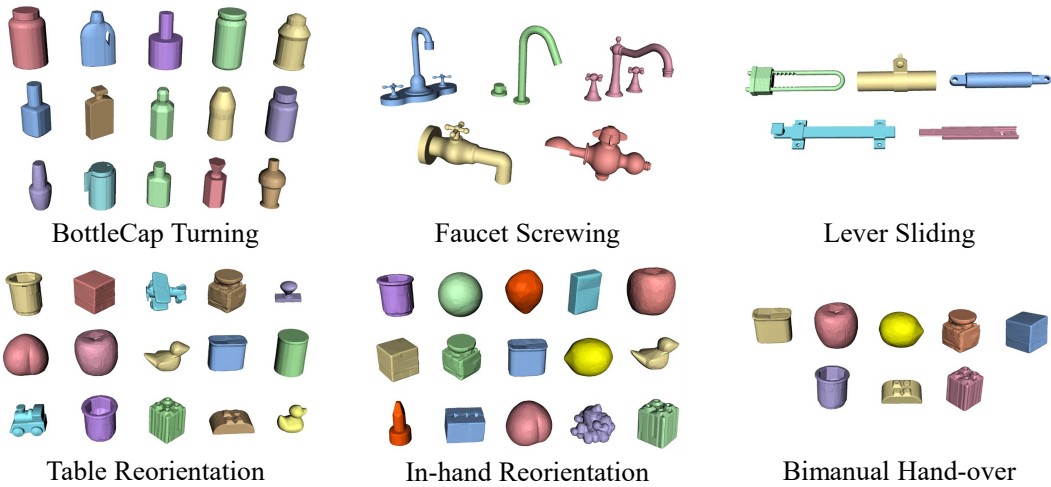

Figure 6: Seen and unseen objects in all tasks. The unseen objects of Faucet Screwing and Lever Sliding are generated by scaling the seen objects with two different scaling factors (0.9 and 1.1)

# A    MORE DETAILS ABOUT THE DATASET

## A.1    TACTILE GLOVE

Our glove system in Fig.8 incorporates 20 commercial piezoresistive pressure sensors as tactile sensing units.  For enhanced sensitivity, we utilize an 18.3 mm diameter sensor on the thumb tip and palm, while the remaining areas are equipped with 10 mm sensors. Each sensor's resistance is converted into voltage signals through a conversion module, which are then processed by an STM32 microcontroller. The system transmits these signals at a rate of 200 Hz to a central computer via a serial port for data storage.

For calibration, we conducted 15 force tests per sensor using a force/torque (F/T) sensor, applying forces ranging from 0.5 N to 7.5 N in 0.5 N increments as shown in Fig.7. Notably, the sliding resistors in the conversion module are fine-tuned to ensure consistent voltage readings across all sensors for the same force. And this calibration is applied to both the left and right-hand gloves. The calibrated force-voltage function is given by:

Figure 7: Force and voltage testing results

$$U = 0.7216 \times F^{0.5025} + 0.0398 \qquad (1)$$

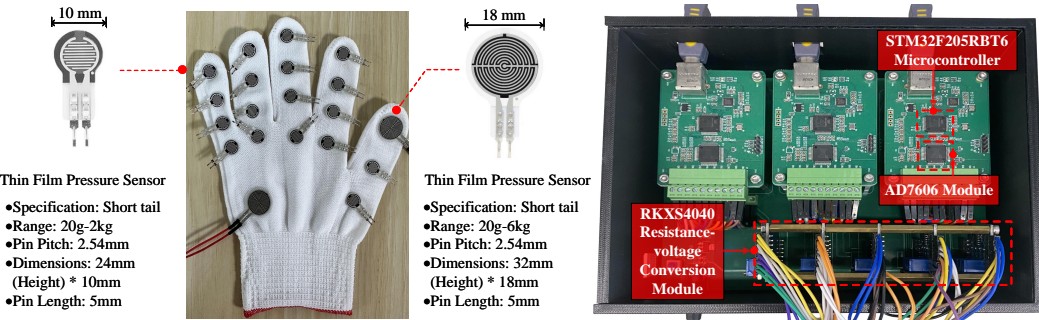

Figure 8: Hardware specification of data collection and glove force and voltage testing results

## A.2    COLLECTION TASKS

Tab.8 shows the task descriptions we collect in our dataset and some demos.

## A.3    T-SNE VISUALIZATION

We set **0.2** (the same as the experiment settings in Sec.B.2) to binarize the tactile voltage signals, converting them into 0-1 contact signals. We perform t-SNE visualization on the binary tactile signals of each trajectory to further investigate the relationship between tactile combination sequences of different operational tasks. Due to the varying length of each operation trajectory and the fact that we rarely care about the relationship between the hand and object before the first contact, we only retain the trajectory between the first and last contact frames for each operation trajectory. According to the length statistics in Fig.1(d), we set the length of each trajectory to 500 frames. For trajectories with insufficient length, we loop the tactile state of the trajectory to reach a length of 500 frames. For any trajectories with more than 500 frames, we directly delete the extra frames.

Table 8: Tasks descriptions and demos in the dataset

| Collected Tasks | Description | # Objects | # Trajs | Examples |
|---|---|---|---|---|
| PickUp | pick up an object and put it down | 50 | 250 | |
| BottleCap Turning | open or close the bottle cap | 30 | 300 | |
| In-hand Reorientation | Rotate the in-hand object with the palm facing up. | 20 | 205 | |
| Bowl Unstacking | unstack a bowl from another | 28 | 269 | |
| Articulated Manipulation | Manipulate the object around the pivot of the hinge. | 10 | 168 | |
| Peg-in-hole | Insert the plug into the socket or Remove the plug from the socket | 8 | 153 | |
| Water Pouring | pour the water into the bowl | 10 | 196 | |
| Table-top Manipulation | manipulate the objects on the table, such as pushing, rotating, flipping, rolling | 10 | 196 | |
| Scissors Manipulation | manipulate tools made using the principle of levers, such as scissors, pliers, and clamps, | 6 | 106 | |
| Pressing | Pressing a button to perform certain functions, such as using a remote control | 10 | 189 | |

## B  DETAILS OF THE BENCHMARK

### B.1  TASK SPECIFICATIONS

In this section, we introduce our visual-tactile dexterous manipulation platform. Due to Isaac Gym Makoviychuk et al. (2021) (BSD 3-Clause License) supporting GPU acceleration and rendering, we construct all the tasks in the Isaac Gym environment.

#### B.1.1  BOTTLECAP TURNING

**Description.**    This task provides a platform for a dexterous hand to learn the skill of rotating bottle caps. Unlike grippers, a dexterous hand can utilize the coordination among its fingers, leveraging lateral friction between the fingers and the cap to induce rotation. In this task, we fix all the bottles on the table, preserving only the rotational freedom of the cap around its own Z-axis. Since rotating a bottle cap with one hand is also challenging for humans, we are more focused on whether the manipulation policy can learn the coordination between the fingers to rotate the cap. The objects used in this task are sourced from the ShapeNet Chang et al. (2015) dataset. Building upon the URDF files of the objects provided in the work Wang et al. (2023) (CC BY-NC 4.0 License), we add a joint with a rotational range of 0 to 6.28 radians to the cap. We prepare 10 objects for skill learning and 5 unseen objects for testing. Using various objects for training allows the learned policy to generalize to unseen bottle caps. This task aims to rotate the bottle cap counterclockwise by one full circle. Achieving this goal requires the dexterous hand to act twisting the cap with fingers in one episode repeatedly.

**Environment settings.**    We place the bottle at the center of the table at coordinates $(0, 0)$, with its height just touching the table surface. We determine the bottle's height by measuring the furthest

distance of the bottle mesh along the Z-axis. The robotic arm is positioned at $(-0.38, -0.02, h+0.05)$, where $h$ is the combined height of the table and the bottle. For this task, the camera parameters are set as eye $= [-0.25, 0.35, 0.5]$ and lookat $= [0.02, -0.02, 0.1]$. The damping of the bottle cap joint is set to 0.1, and all other simulation parameters use default values.

### B.1.2 FAUCET SCREWING.

**Description.** Faucet Screwing is similar to BottleCap Turning. However, when constructing the task, we introduce distinct designs to ensure the diversity of the tasks. We add taps with handles to the task so that the hand needs to push the handle with fingers. Additionally, we define the goal of the task as rotating the tap handle one full circle clockwise, necessitating different finger gaits compared with the goal of BottleCap Turning. We choose five faucet models with a rotational axis perpendicular to the ground from the SAPIEN Xiang et al. (2020) (MIT license) dataset for manipulation policy learning and generate 10 test objects by scaling the taps using two different scaling factors (1.1 and 0.9) to evaluate the generalization of the learned policy. We limit the rotation range of the faucet to 0-6.28 radians.

**Environment settings.** To place all the faucets on the table, we scale, rotate and adjust the height of different objects. Then, we rotate the robotic arm using Euler angles (3.14, 0, 1.57) in the ZYX order and place the objects in different positions for each faucet. The specific settings for these adjustments are detailed in Tab.9. We set the damping for all faucet joints to 0.1 and the friction to 0.5. The mass of each faucet is set to 0.15 kg. For this task, the camera parameters are set as eye $= [0.4, -0.4, 0.5]$ and lookat $= [0, 0, 0.05]$.

Table 9: Hand position for different objects

| Obj ID | Scale | Height | Euler(zyx) | Hand position |
|--------|-------|--------|------------|---------------|
| 886 | 0.3 | 0.1 | (0, 0, 3.14) | (0.32, 0.12, 0.72) |
| 1386 | 0.3 | 0.07 | (0, 0, 3.14) | (0.38, 0.12, 0.74) |
| 2017 | 0.25 | 0.0 | (0, 0, 3.14) | (0.36, -0.01, 0.75) |
| 2095 | 0.15 | 0.02 | (0, 1.57, 0) | (0.37, -0.015, 0.73) |
| 2113 | 0.3 | 0.2 | (0, 0, 1.57) | (0, 1.57, 0) |

### B.1.3 LEVER SLIDING.

**Description.** The lever sliding task focuses on actions to separate the long hole and the axe. Without the assistance of arm movement, this task requires the fingers to press or pinch the object and use wrist motion to slide the axis out. Unlike the previous two tasks, which involve complex coordination between fingers, this task is a simple coordinated movement where all fingers perform the same motion pattern. However, it places greater demands on the coordination between wrist joint and finger movements. In this task, we select five common long-hole and axis combinations in daily life, including (1) ClampingHanger, (2) DeadboltLock, (3) GasSpring, (4) Padlock, and (5) TelescopingSlide. We create the CAD models and manually generate the corresponding URDF files. Due to the varying lengths of the holes in each model, we uniformly define the completion goal for all models in this task as sliding the axis out by 15 $cm$. We train with the five objects and, like the Faucet Screwing task, create 10 test objects by scaling in two sets (1.1 and 0.9).

**Environment settings.** In this task, we rotate the robotic arm using Euler angles (3.14, 0, 1.57) in the ZYX order and the scale, positions and rotations of the objects and the hand position are shown in Tab.10. We set the force and velocity limits for the sliding joints to 1, the joint stiffness to 1, damping to 0.5, and joint friction to 1.5. We set the surface friction coefficient of the objects to 1 and their mass to 1 kg. For this task, the camera parameters are set as eye $= [0.3, -0.25, 0.3]$ and lookat $= [0, 0, 0]$.

Table 10: Object settings and hand positions in the Lever Sliding task.

| Obj ID | Scale | Position | Euler(zyx) | Hand position |
|--------|-------|----------|------------|---------------|
| 1 | 0.25 | (0, 0, 0.63) | (1.57, 0, 0) | (0.4, 0.1, 0.71) |
| 2 | 0.2 | (0, 0, 0.67) | (-1.57, 0, -1.57) | (0.4, 0.08, 0.71) |
| 3 | 0.12 | (0, -0.12, 0.65) | (-1.57, 0, 0) | (0.4, 0.13, 0.71) |
| 4 | 0.15 | (0, -0.12, 0.65) | (0, 1.57, 0) | (0.38, 0.08, 0.71) |
| 5 | 0.15 | (0, 0, 0.66) | (1.57, -1.57, 0) | (0.4, 0.08, 0.72) |

### B.1.4 TABLE REORIENTATION

**Description.** Sometimes, when we want to pick up an object from the table, we aim to position it appropriately before grasping it. In this task, the dexterous hand attempts to rotate the object on the table while ensuring that it does not topple over. This task rigorously tests the coordination between

the thumb and the other four fingers, as the object is prone to tipping due to uneven forces. The goal of this task is to achieve a half-circle rotation of the object while keeping it stable. To ensure that the skills learned by the dexterous hand can generalize to different objects, we selected 10 objects from the YCB dataset for training and chose 5 objects for testing.

**Environment settings.** In this environment, the dexterous hand rotates an object on the table. We set the table height at 0.6 meters and positioned the robotic hand at (-0.03, 0.4, 0.72), rotated 180 degrees around the Y-axis. The object is initialized at (0, 0.02, 0.67) and freely falls onto the table after the round begins. All objects have a friction coefficient of 0.8, and their RGB color is set to (204, 204, 0).

### B.1.5 IN-HAND REORIENTATION.

**Description.** The in-hand reorientation task involves rotating an object with the fingers while keeping the palm facing upward. The goal is to rotate the object anticlockwise over a large angle without deviating beyond a specified threshold. This task adds complexity compared to previous ones, as maintaining an upward-facing palm during rotation increases the risk of the object falling if not properly stabilized. Rotations over half a circle without exceeding the deviation threshold along the Z-axis from the target position are considered successful. Due to the frequent changes in the object's Z-axis position, the fall condition is relaxed, allowing for a small amount of Z-axis bias. 10 objects from the YCB dataset are used during training, and 5 are used for testing.

**Environment settings.** Since the table is not involved in this task, it has been removed from the simulation environment. The robot arm is initially positioned at coordinates (0, 0, 0.5), and the object is placed at (0, -0.39, 0.56), which aligns it with the center of the robot's palm. The surface friction coefficient between the objects and the environment is set to 0.8 to simulate realistic contact interactions. The masses of the objects are randomly assigned within the range of 0.5kg to 1.5kg to introduce variability in weight. For this task, the camera parameters are configured with the eye position at $[0.15, 0.15, 0.165]$ and the look-at point at $[0.1, 0, 0.05]$.

### B.1.6 BIMANUAL HAND-OVER

**Description.** Unlike previous tasks, this is a bimanual manipulation task designed to teach the dexterous hands the skill of coordination. In this task, one hand throws the object while the other hand needs to catch it and ensure it does not drop. This requires the dexterous hand to coordinate finger joints and wrist joints, allowing the object to be thrown in the correct direction and at the appropriate speed. The catching hand must use its various joints to counteract this speed and stabilize the object in its grasp. This task includes 8 objects for training and 5 objects for testing.

**Environment settings.** The task environment consists of two Shadow Hands and an object. The throwing hand is positioned at (0, 0, 0.5), while the catching hand is placed at (0, -1, 0.5) and rotated (0, 0, 3.1415) radians in the ZYX order. All objects are initialized at (0, -0.39, 0.54), directly above the throwing hand. Each object's mass is set to 0.1 kg, with RGB color set to (204, 204, 0). The target position for the object is (0, -0.64, 0.54), and the task is considered successful when the catching hand brings the object close to this target position.

### B.2 DETAILS OF REPRESENTATION LEARNING METHODS

#### B.2.1 VISUAL-TACTILE FUSION MODEL

Inspired by MAE (masked autoencoder) He et al. (2022), we construct a benchmark method to fuse visual and tactile modalities, which consists of two parts: the fusion encoder and the reconstruction decoder. Fig.9 shows the framework of our fusion algorithm.

**Fusion Encoder** The fusion encoder $E_\theta$ processes image-tactile data pairs $(V, C)$ to output a visual-tactile fused representation $h$, which encompasses three stages: extraction (Eq.2), masking (Eq.3), and fusion (Eq.4).

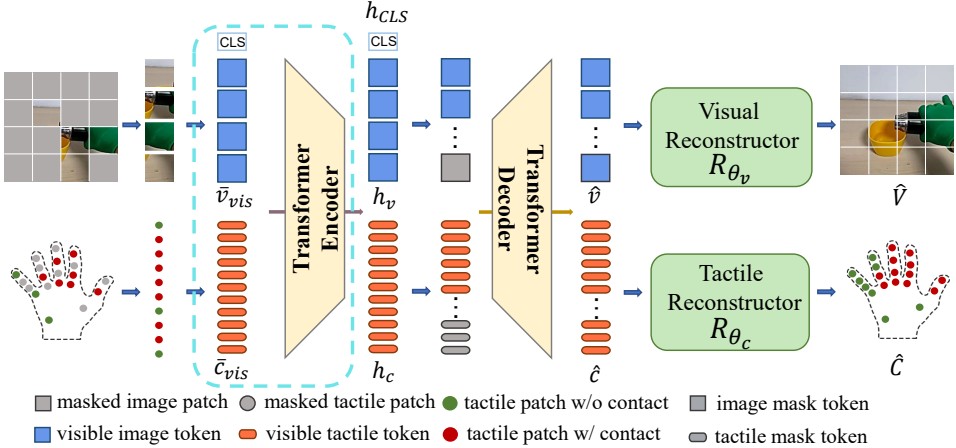

Figure 9: The network structure of VT-JointPretrain

For **extraction**, the input RGB image $V \in \mathbb{R}^{H \times W \times 3}$ is divided and flattened to $v \in \mathbb{R}^{N_v \times (P^2 \cdot 3)}$, where $N_v$ is the number of image patches and $(P, P)$ is the resolution of each patch. And we apply a linear projection $\phi_\theta(\cdot)$ to image patches $v$ and add 2D sine-cosine positional encoding $v^{pos}$ to generate image patches embeddings $\bar{v} \in \mathbb{R}^{N_i \times d_{en}}$. The input tactile $C \in \{0, 1\}^{N_c}$ is sliced into $N_c$ tactile patches, each of which is projected with an MLP layer $\varphi_\theta(\cdot)$ and add 1D sine-cosine positional encodings $c^{pos}$ to produce tactile patch embeddings $\bar{c} \in \mathbb{R}^{N_c \times d_{en}}$. $d_{en}$ is the dimensionality of the encoder output. The extraction is described as:

$$\bar{v} = \phi_\theta(v) + v^{pos}, \bar{c} = \varphi_\theta(c) + c^{pos} \tag{2}$$

For **masking**, we introduce a modality-specific masking function denoted as $M(\cdot, \gamma)$, where $\gamma$ represents the masking ratio for the respective input modality. Utilizing this function, we apply masking to the image (or tactile) patch embeddings with a given mask ratio $\gamma_v$ (or $\gamma_c$). This results in the output of visible patch embeddings $\bar{v}_{vis} \in \mathbb{R}^{(1-\gamma_v)N_v \times d_{en}}$ (or $\bar{c}_{vis} \in \mathbb{R}^{(1-\gamma_c)N_c \times d_{en}}$). We specify the masking process as follows:

$$\bar{v}_{vis} = M(\bar{v}, \gamma_v), \bar{c}_{vis} = M(\bar{c}, \gamma_c) \tag{3}$$

For **fusion**, the main architecture is a transformer encoder $TransE(\cdot)$. We concatenate a learnable CLS token with visiable patched embeddings $(\bar{v}_{vis}, \bar{c}_{vis})$. The visual-tactile fusion representation $h \in \mathbb{R}^{s \times d} = \{h_{CLS}, h_v, h_c\}$ can be defined as:

$$h = TransE(CLS, \bar{v}_{vis}, \bar{c}_{vis}) \tag{4}$$

where $s = (1 - \gamma_v)N_v + (1 - \gamma_c)N_c + 1$ and $d$ is the dimensionality of the fusion representation. $h_{CLS}, h_v$ and $h_c$ represent latent features of each input.

**Reconstruction Decoder** Following MAE, the reconstruction decoder $D_\theta$ is responsible for reconstructing masked patches. Firstly, $h_v, h_c$ and mask tokens $m \in \mathbb{R}^{\gamma_v \times N_v + \gamma_c \times N_c}$ are fed into a transformer decoder $TransD(\cdot)$, represented as follow:

$$\{\hat{v}, \hat{c}\} = TransD(h_v, h_c, m) \tag{5}$$

where $\hat{v} \in \mathbb{R}^{N_i \times d_{de}}$ (or $\hat{c} \in \mathbb{R}^{N_c \times d_{de}}$) is the reconstructed image (or tactile) patch embeddings. $d_{de}$ is the dimensionality of the reconstructed patch embeddings.

We use an MLP as the visual reconstructor $R_{\theta_v}$, which projects $\hat{v}$ into $\hat{V} \in \mathbb{R}^{H \times W \times 3}$. The tactile reconstructor $R_{\theta_c}$ is an ensemble of Multi-Layer Perceptrons (MLPs), where each MLP is utilized to individually map reconstructed tactile patch $\hat{c}_i \in \hat{c} = \{\hat{c}_0, \hat{c}_1, \dots, \hat{c}_{N_c}\}$ to their reconstructed representations $\hat{C}_i \in \mathbb{R}^1$. Thus, the reconstructors are formulated as follows:

$$\hat{V} = R_{\theta_v}(\hat{v}), \hat{C} = R_{\theta_c}(\hat{c}) \tag{6}$$

Table 12: Visual pretrained models used in our benchmark

| Pretrained models | Method | Architectures |
|---|---|---|
| CLIP Radford et al. (2021) | Contrastive learning Hadsell et al. (2006) | ViT-B/16 |
| R3M Nair et al. (2022) | Time contrastive learning Sermanet et al. (2018) | Resnet-18 |
| MVP Radosavovic et al. (2023) | Masked autoencoder He et al. (2022) | ViT-Base |
| Voltron Karamcheti et al. (2023) | Multimodal masked autoencoder Geng et al. (2022) | ViT-Small |
| ResNet18 He et al. (2016) | Supervised learning | Resnet-18 |

**Loss Function**  Our visual-tactile representation learning algorithm leverages the reconstruction of invisible patches from visible ones, enabling the integration and interaction of information between modalities. This process facilitates the learning of semantic associations both within and between modalities. To evaluate the quality of the reconstruction, we define the following loss function:

$$L(\theta) = \lambda_v \cdot \text{MSE}(V, \hat{V}) + \lambda_c \cdot \text{MSE}(C, \hat{C}) \tag{7}$$

where $\lambda_v$ and $\lambda_c$ are the loss weight for vision and tactile modality.

**Dataset preprocessing.**  We first trim the first and last 10% of each operation sequence, retaining the data pairs that include activated tactile signals. Then, we split the dataset into training and test sets with an 8/2 ratio. The images in the dataset have a resolution of 420x240. Before feeding the images into the network, we perform a center crop and resize to a resolution of 224x224. The tactile data is binarized by a threshold (0.2). We feed the processed visual-tactile data pairs into the network for training and testing.

Table 11: The hyperparameters of our visual-tactile fusion model

| Hyperparameters | All Models |
|---|---|
| Image resolution $(H, W)$ | (224, 224) |
| Num image patches $N_v$ | 196 |
| Num tactile patches $N_c$ | 20 |
| Patch resolution $(P, P)$ | (16, 16) |
| Encoder input dim $d_{en}$ | 384 |
| Image mask ratio $\gamma_v$ | 0.75 |
| Tactile mask ratio $\gamma_c$ | 0.5 |
| Fusion dim $d$ | 384 |
| Decoder output dim $d_{de}$ | 192 |
| Image loss weight $\lambda_v$ | 1 |
| Tactile loss weight $\lambda_c$ | 10 |
| Learning rate | 1.5e-4 |
| Batch size | 64 |

**Training details**  Tab.11 lists the hyperparameters used during our training. For parameters not listed, we followed the MAE settings. We trained on our data for 250 epochs in parallel on two NVIDIA GTX 3090 GPUs, which took approximately 25 hours. We selected the model from the 210th epoch for downstream visual-tactile representation extraction. The other two single-modality pretrained methods using our data (V-Pretrain and T-Pretrain) also used these parameters but selected models from the 170th and 310th epochs, respectively.

### B.2.2 OTHER PRETRAINED METHODS

We utilize the open-sourced models from these works directly, with the corresponding model names being "resnet18", "vitb-mae-egosoup", "ViT-B/16", and "v-cond+vit-small+sth-sth+epoch-400". The models we use are listed in Tab.12.

**CLIP Radford et al. (2021).**  The core of the **CLIP** model is to train a model that understands the relationship between image content and textual descriptions through a contrastive learning approach. It is pre-trained on a large-scale dataset of image and text pairs, learning to map visual and text information into a common feature space. We import the CLIP model into our RL framework by calling `clip.load("ViT-B/16", device=device)`, which is provided on the Github[3].

**R3M Nair et al. (2022).**  **R3M** employs two contrastive learning objectives: time contrastive learning and image-language temporal alignment. It leverages the temporal structure of videos, aiming to maximize similarity between adjacent frames while contrasting frames that are further apart. Additionally, R3M utilizes language supervision, combining language descriptions with dual-frame contexts to capture task progression. We import the R3M model into our RL framework by calling `r3m = load_r3m("resnet18")`, which is provided on the Github[4].

**MVP Radosavovic et al. (2023).**  **MVP** employs the concept of masked autoencoders He et al. (2022), where parts of the input image are masked, and then a transformer is used to integrate and

---

[3]`https://github.com/openai/CLIP`
[4]`https://github.com/facebookresearch/r3m`

reconstruct the image. We import the MVP model into our RL framework by calling `model = mvp.load("vitb-mae-egosoup")`, which is provided on the Github[5].

**Voltron Karamcheti et al. (2023).** **Voltron** is a framework for language-driven representation learning from human videos and associated captions. It balances between language-conditioned visual reconstruction to help in learning low-level visual patterns, and visually-grounded language generation, which encodes high-level semantics. We import the MVP model into our RL framework by calling `voltron.load("v-cond", device="cuda", freeze=True)`, which is provided on the Github[6].

**ResNet18 He et al. (2016).** **ResNet18** is trained with ImageNet-1k Krizhevsky et al. (2012) by supervised learning. We import the ResNet18 model into our RL framework by calling `torchvision.models. resnet18(pretrained=True)`. When using the model, we removed the final linear layer.

**Feature extraction in RL.** For all pretrained models, we retain only the visual feature extractor and freeze all parameters. When using models for the visual modality alone (such as **V_CLIP**), the extracted visual features are combined with proprioceptive information and feed into the policy network. For models using visual-tactile modalities (such as **V_CLIP+T**), an additional tactile feature extracted by an MLP is also input into the network.

### B.2.3 NON-PRETRINED METHODS

We use the network structure of ResNet18 to extract image features and an MLP to extract tactile features, forming three non-pretrained baseline models: **T**, **V** and **V+T**. We use `torchvision.models.resnet18()` to load the model.

### B.3 RL FRAMEWORK

### B.3.1 RL MODELING

We model the dexterous manipulation task as a Markov Decision Process (MDP), defined by a tuple: $(\mathcal{S}, \mathcal{A}, \mathcal{T}, \mathcal{R}, \gamma)$. $\mathcal{S}$ and $\mathcal{A}$ represent the state and action space. The policy $\pi_\theta : \mathcal{S} \to \mathcal{A}$ maps the state space $\mathcal{S}$ to the action space $\mathcal{A}$. $\mathcal{T} : \mathcal{S} \times \mathcal{A} \to \mathcal{S}$ is the transition dynamic. $\mathcal{R} : \mathcal{S} \times \mathcal{A} \to \mathbb{R}$ is the reward function and $\gamma \in (0, 1]$ is the discount factor. Our goal is to maximize the expected discounted reward $J(\pi) = \mathbb{E}_\pi \left[ \sum_{t=0} \gamma^t r(s_t, a_t) \right]$ to train a policy network $\pi_\theta$. We use the PPO Schulman et al. (2017) algorithm to make the agent learn manipulation skills.

**State Space.** In all tasks, we define the state as $S = \{h \leftarrow \mathcal{M}_\theta(\cdot), \mathcal{P}\}$. $M_\theta(\cdot)$ take the RGB image $V_{sim}$ or the binarized tactile signals $C_{sim}$ as input and generate the perceptual representation $h$. $V_{sim}$ is captured by the ego-centric camera in the simulator and $C_{sim}$ is the binarized result of the signal obtained from the force sensors, with a tactile threshold set to 0.01 N across all tasks. $\mathcal{P}$ represents the proprioceptive information of the dexterous hand, which includes the joint angles and joint velocities of the hand. We do not incorporate the extra information of the dexterous hand intentionally, primarily to encourage the manipulation strategy to focus more on visual-tactile perceptual signals. Moreover, the joint angles and joint velocities of the hand are readily accessible in real-world environments.

**Action Space.** In all tasks, we utilize the Shadow Hand as the operator, which possesses 24 degrees of freedom. However, four of these joints are actuated through tendons. We immobilize the arm of the Shadow Hand, allowing it to complete each task solely with the motion of its fingers. Consequently, in each task, the action $a \in \mathbb{R}^{20}$.

**Reward Design of BottleCap Turning.** We define the reward function as:

$$r = \lambda_1 r_p + \lambda_2 r_v + \lambda_3 r_d + \lambda_4 r_s \tag{8}$$

---

[5] `https://github.com/ir413/mvp`
[6] `https://github.com/siddk/voltron-robotics/tree/main`

where the position reward $r_p = min(\theta_{joint}, 7.0)$, the velocity reward $r_v = clamp(v_{joint}, -10, 10)$, the distance reward $r_d = \exp^{-10d}$ and the success reward $r_s = 5$. $d$ represents the sum of the distances from each fingertip to 2 centimeters below the bottle cap. We set $\lambda_1 = 0.5$, $\lambda_2 = 1$, $\lambda_3 = 0.5$ and $\lambda_4 = 1$. Especially, if $r_v > 0$ but the tactile sensors are not activated, then $\lambda_2$ will be set to 0 when calculating the reward, aiming to encourage the dexterous hand to use the part of the tactile sensor that makes contact with the bottle cap to generate positive rotation.

**Reward Design of Faucet Screwing.** The reward function for the faucet is similar to that for the bottle cap, with the difference being that the contact condition is not considered when calculating the velocity reward. Additionally, we set a target height for each faucet that we aim for the fingers to reach. In the Z-axis direction, these target heights are offset from the faucet positions by [-0.02, -0.01, 0.08, -0.02, -0.01]. $d$ is the sum of the distances from each fingertip to these positions.

**Reward Design of Lever Sliding.** We define the reward function as:

$$r = \lambda_1 r_p + \lambda_2 r_v + \lambda_3 r_d \tag{9}$$

where the position reward $r_p = \theta_{joint}$, the velocity reward $r_v = v_{joint}$ and the distance reward $r_d = \exp^{-10d_1} + \exp^{-10d_2}$. We set the target position 8 cm away from the center of the object in the direction of the axis coming out. The $d_1$ is the sum of the distances between the fingertips and the target position. The $d_2$ is the distance between the palm and the target position. We set $\lambda_1 = 1$, $\lambda_2 = 2$, $\lambda_3 = 1$.

**Reward Design of Table Reorientation.** We define the reward function of this task as:

$$r = \lambda_1 r_d + \lambda_2 r_{rot} + \lambda_3 r_a + \lambda_4 r_v + r_b \tag{10}$$

where $r_d = \left[\exp^{-10d_1}, d_2\right]$ contains the distance $d_1$ between the fingertips and the object along the z-axis and the Euclidean distance $d_2$ to the target in the xy-plane. $r_{rot}$ represents the difference between the current pose and the target pose of the object. $r_a = ||a||_2$ is the action reward and $r_v = clamp(\omega_z, -10, 10)$ is the velocity reward for the z-axis rotation. $r_b = 250$ is the bonus when the agent achieves the goal. We set the $\lambda_1 = \begin{bmatrix} 0.25 \\ -10 \end{bmatrix}$, $\lambda_2 = 1$, $\lambda_3 = -0.0002$ and $\lambda_4 = 1$.

**Reward Design of In-hand Reorientation.** The reward function is defined as:

$$r = \lambda_1 r_d + \lambda_2 r_{rot} + \lambda_3 r_a + \lambda_4 r_v \tag{11}$$

where the distance reward $r_d = \left[\exp^{-10d_1}, d_2\right]$, the absolute rotation $r_{rot}$, the action reward $r_a$ and the velocity reward $r_v = clamp(\omega_z, -10, 10)$. $d_1$ represents the distance between the fingertips and the object along the z-axis and $d_2$ is the Euclidean distance to the target in the xy-plane. We set $\lambda_1 = \begin{bmatrix} 0.25 \\ -10 \end{bmatrix}$, $\lambda_2 = 1$, $\lambda_3 = -0.0002$ and $\lambda_4 = 1$.

**Reward Design of Bimanual Hand-over.** The reward function of Bimanual Hand-over is represented as:

$$r = r_d + r_b \tag{12}$$

where $r_d = exp^{-0.2*(d_1*50+d_2)}$. $d_1$ is the Euclidean distance between the current position and the target position and $d_2$ is the distance between the current pose and the target pose of the object. $r_b = 250$ is the bonus reward.

### B.4 TRAINING DETAILS

For all tasks, we set up 200 environments in Isaac Gym Makoviychuk et al. (2021) to collect trajectories in parallel. The maximum episode length is 500 for BottleCap Turning and Faucet Screwing, 250 for Lever Sliding and 600 for other three tasks. Upon each reset, the fixed position of the robotic arm is perturbed by 1 cm, each joint angle is reset to a random value within -0.05 to 0.05 rad, and the joint velocities are reset to a random value within -0.1 to 0.1 rad/s.

In each step, the camera captures a 224×224 image, or the pressure values from 20 tactile sensors are binarized using a threshold of 0.01 N. This sensory information is fed into the representation

extraction model to obtain the representation information. A linear layer maps this representation to a 128-dimensional vector, which is concatenated with another 128-dimensional vector representing proprioceptive information, also mapped through a linear layer. This combined information is then fed into the policy network $\pi_\theta$.

Our policy network has hidden layer sizes of [1024, 1024, 512] and uses ELU Clevert et al. (2015) as the activation function. The policy network is optimized using the PPO Schulman et al. (2017) algorithm. Tab.13 shows the hyperparameters of the PPO algorithm. Although the handover task involves two hands, we do not use multi-agent reinforcement learning Wang et al. (2024) to train them. Instead, the policy network directly outputs the actions for both hands.

We run all experiments for 2000 or 3000 iterations on a device equipped with an Intel Xeon Gold 6326 processor and an NVIDIA 3090 GPU. For methods using both vision and tactile input, training the manipulation policy takes approximately 14 or 21 hours. For methods using only vision input, it takes about 10 or 15 hours to train the visual manipulation policy. For methods using only tactile input, training the manipulation policy takes about 4 or 6 hours. However, for methods without pretraining that include vision, training for 2000 iterations can take up to a week.

Table 13: The hyperparameters of PPO

| Hyperparameters | All Tasks |
|---|---|
| Num mini-batches | 4 |
| Num opt-epochs | 10 |
| Rollout step | 32 |
| Hidden size | [1024, 1024, 512] |
| Activation | ELU |
| Clip range | 0.2 |
| Max grad norm | 1 |
| Learning rate | 3.e-4 |
| Discount ($\lambda$) | 0.96 |
| GAE lambda ($\gamma$) | 0.95 |
| Init noise std | 0.8 |
| Desired kl | 0.016 |
| Ent-coef | 0 |

### B.5 VISUALIZAITON OF MANIPULATION POLICES

Fig. 5 visualizes the results of RL policy training for different tasks and objects on our manipulation platform.

### B.6 DETAILS OF VIEWPOINT ADAPTABILITY.

Tab.14 represents the 3 different view settings.

Table 14: Viewpoint settings across three tasks

| Task | Viewpoint | Eye | Lookat |
|---|---|---|---|
| BottleCap Turning | ego-centric | (-0.25, 0.35, 0.5) | (0.02, -0.02, 0.1) |
| | on the arm | (-0.25, 0.0, 0.5) | (0.02, 0, 0.1) |
| | third view | (0.25, 0.35, 0.5) | (-0.02, -0.02, 0.1) |

## C REAL WORLD EXPERIEMNTS

### C.1 SYSTEM SETUP

In order to evaluate the effectiveness of the trained policy, we established a system comprising a Shadow Hand Sharma et al. (2014) and an Azure Kinect camera[7]. To collect tactile data, we attach 20 piezoresistive tactile sensors to each part of the Shadow Hand. We calibrate the tactile sensors and collect the tactile signals using the same system, described in A.1. We set the tactile threshold to 0.2V for the dexterous manipulation platform. In all tasks, the position of the arm remains fixed.

### C.2 SIM2REAL PARAMETERS FOR DIFFERENT MODALITIES

We train a robust policy in simulation by first applying domain randomization to proprioceptive data and introducing tactile noise. Next, we distill the trained policy into a student policy to handle the image sim2real transfer.

---

[7]https://azure.microsoft.com/en-us/products/kinect-dk

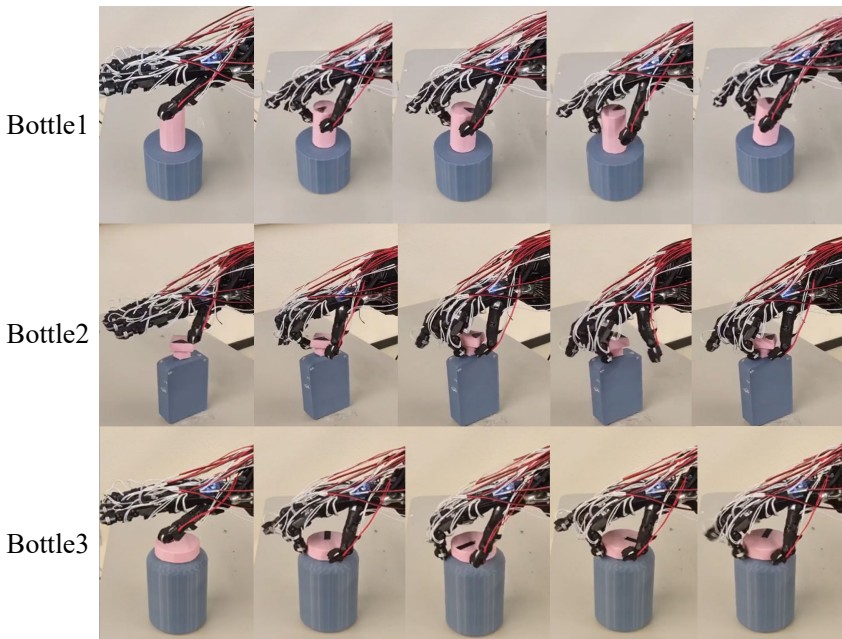

Figure 10: More BottleCap Turning real experiments with different objects.

**For proprioceptive information and actions**, there are 24 dimensions of joint angles and 24 dimensions of joint velocities. We apply domain randomization Tobin et al. (2017) to both the proprioceptive data and the actions. The parameters are shown in Tab.15.

Table 15: Real-world experiment settings

|  | Joint angles & Joint vel (48dim) | Actions (20dim) |
|---|---|---|
| Noise (mean, std) | [ 0, 0.002] | [ 0, 0.05] |
| **Correlated noise (mean, std)** | [ 0, 0.001] | [ 0, 0.015] |
| Operation | Additive | Additive |
| Distribution | Gaussian | Gaussian |
| Schedule | Linear | Linear |
| Schedule_steps | 40000 | 40000 |

**For RGB images**, we employ various image augmentation techniques and the color of the manipulative objects on real RGB images to address the sim2real problem:

- **Pixel randomization.** We add uniform noise in the range of [-5, 5] to each pixel independently and pixels of values outside the range of [0, 255] are clipped

- **Random perspective transformation.** We uniformly randomize the offset of the four corners of the source image with the range of [0, 30].

- **Contrast and lightness randomization.** The contrast parameter is randomly generated in the range of [-20, 20] and the lightness parameter is randomly generated in the range of [-30, 30].

- **Manipulated object color settings.** We set different colors for various objects.

**For tactile**, we introduce Gaussian noise with standard deviations of 0.1N to the tactile signals, keeping the binarization threshold at 0.01N as **v3** in 4) of Sec.5.3 dose.

### C.3 DEPLOYMENT THE ROBUST POLICY IN REAL WORLD

We use ROS to obtain the joint angles and velocities of the Shadow Hand, Kinect to capture the visual information of the scene, and tactile sensors attached to the hand to collect tactile signals. A Kinect camera is manually calibrated to match the position and orientation of the simulated camera

and we crop the RGB images with a size of 224×224×3, which matches the pixel observation size in the simulation. The tactile sensors capture voltage signals, which are binarized using a 0.2V threshold. All real-world observations are fed into the trained policy, which outputs actions to control the movement of the robot hand.

Due to hardware limitations, we only conducted single-hand tasks in the real-world environment. The qualitative results are shown in Fig.4 and more real experiments about BottleCap Turning can be seen in Fig.10. For demos, see the supplementary video in the zip file.

Tab.16 provides quantitative results of VT-JointPretrain for each task, with two objects per task. A trial is considered successful if the goal is achieved within 15 seconds. Trials exceeding the time limit or resulting in object drops are counted as failures. Each object is tested 10 times.

Table 16: Quantitative results of real-world experiments. We conducted 20 trials for each task.

| Tasks | BottleCap Turning | Faucet Screwing | Lever Sliding | Table Reorientation | In-hand Reorientation |
|---|---|---|---|---|---|
| Succ Times | 16 | 14 | 15 | 13 | 10 |

