# OpenReview forum: "VTDexManip: A Dataset and Benchmark for Visual-tactile Pretraining and Dexterous Manipulation with Reinforcement Learning"
_ICLR.cc/2025/Conference — ICLR 2025 Poster_

### Official Review · Reviewer_Z9Mf · 2024-10-28

**Soundness:** 3
**Presentation:** 3
**Contribution:** 3
**Rating:** 6
**Confidence:** 5

**Summary:**

This paper introduces a dataset and benchmark to address the limitations in current robotic manipulation learning frameworks by incorporating both vision and tactile modalities. It collects human manipulation data with 182 objects and 10 daily tasks and uses low-cost tactile gloves to create a large-scale dataset. The dataset is used to study multi-modal pretraining and dexterous robotic manipulation using reinforcement learning. The authors benchmark 17 pretraining and non-pretraining methods to evaluate the effectiveness of visual and tactile inputs in dexterous manipulation tasks.

**Strengths:**

1. This paper create the first large-scale visual-tactile dataset for dexterous manipulation tasks, addressing a gap in the existing robotic learning literature. The dataset covers a wide variety of tasks and objects, which makes it valuable for generalizing robotic manipulation skills.

2. The use of both vision and tactile information for joint pretraining boosts performance in complex dexterous tasks.

3. The benchmark evaluates various pretraining and non-pretraining methods, which provides a comprehensive understanding of how different modalities affect task performance.

**Weaknesses:**

1. The contribution to the community is not super clear to me. Currently, what kind of tactile sensor should be used on the Dexterous Hand is still an open question. Although this paper discuss the Gelsight in the section 5.5, there are still a lot of different sensors. Optical tactile sensor [1, 6] has been used for dexterous manipulation, where "they have not been evaluated for dexterous manipulation with complex dynamics and coordination between fingers" shown in the paper might be incorrect. Also, magnetic tactile sensor [2, 3], force tactile sensor [4] are used in dexterous manipulation tasks. It's not clear why this type of tactile data is chosen for pre-training. Why does this benchmark could be widely used since it's still an open question?

2. There are some paper uses visual-tactile representations for reinforcement learning [5, 6]. I think compare using the pre-trained with those SOTA online-learning model is helpful (not only using simple ResNet and MLP).

3. Section 5.4 shows that this framework can do Sim2Real and show some impressive demos. However, no quantitative results have been shown in the paper.

4. A new CoRL paper [7] present a method to learn tactile-guided control policy from human hand demonstrations by embedding tactile sensor on the human finger tip, where the idea is similar to collect human-tactile data with tactile glove shown in this paper. What's the difference between these two approaches? It's helpful if analyzing it in related work.

5. In section 5.2, the paper didn't show which task is it.

[1]. Lambeta et al., DIGIT: A Novel Design for a Low-Cost Compact High-Resolution Tactile Sensor With Application to In-Hand Manipulation, 2020

[2]. Bhirangi et al., ReSkin: a versatile, replaceable, low-cost skin for AI research on tactile perception, 2021

[3]. Bhirangi et al., AnySkin: Plug-and-play Skin Sensing for Robotic Touch, 2024

[4]. Yin et al., Rotating without Seeing: Towards In-hand Dexterity through Touch, 2023

[5]. Sferrazza et al., The Power of the Senses: Generalizable Manipulation from Vision and Touch through Masked Multimodal Learning, 2024

[6]. Qi et al., General In-Hand Object Rotation with Vision and Touch, 2023

[7]. Yu et al., MimicTouch: Leveraging Multi-modal Human Tactile Demonstrations for Contact-rich Manipulation, 2024

**Questions:**

1. This paper only compare the VT-JointPretrain with V-Pretrain, V-Pretrain+T, T-Pretrain, and V+T. How about using V-Pretrain+T-Pretrain separately?

More questions have been shown in Weakness. I will consider increasing my score if the contribution can be explained more clearly.

---

> ### Author Response · Authors · 2024-11-22
> **Authors' response to Reviewer Z9Mf (1/3)**
>
> We sincerely appreciate your insightful questions and suggestions. Our work introduces the first dataset designed for multi-finger dexterous manipulation and establishes a comprehensive benchmark featuring six complex manipulation tasks and 17 baseline methods built on existing approaches. We are delighted to engage in a discussion about the selection of tactile sensors for dexterous hand manipulation. Regarding your question about the broader applicability of our benchmark, we have provided a detailed explanation to address this concern. Additionally, for your other questions, we have included extensive experimental data. All our responses are provided below.
>
> >The contribution to the community is not super clear to me. Currently, what kind of tactile sensor should be used on the Dexterous Hand is still an open question. Although this paper discuss the Gelsight in the section 5.5, there are still a lot of different sensors. Optical tactile sensor [1, 6] has been used for dexterous manipulation, where "they have not been evaluated for dexterous manipulation with complex dynamics and coordination between fingers" shown in the paper might be incorrect. Also, magnetic tactile sensor [2, 3], force tactile sensor [4] are used in dexterous manipulation tasks. It's not clear why this type of tactile data is chosen for pre-training. Why does this benchmark could be widely used since it's still an open question?
>
>
>
> **Response to weakness 1:** Thank you for raising this important point. Indeed, there is a wide variety of tactile sensors in the robotics field, each widely used in different applications. Our goal is to explore whether vision-tactile demonstrations from human demonstrations can facilitate the learning of complex manipulation skills for dexterous hands. However, **collecting tactile data from human hands during complex operations poses significant challenges:**
>
> - the tactile sensors cannot be worn out easily;
> - they should be flexible with hand movements ;
> - thin and easy to stick to hands;
> - compact when put together into a glove or a device;
> - easy for others to reproduce our collection;
> - low-cost and readily available for many researchers not specializing in tactile sensors;
> - if they do not provide high precision or high-resolution tactile signals, they should provide some basic features for the tactile signals to start the investigation, like contact or not, how large the force of the contact.
>
> Piezoresistive tactile sensors meet all the requirements. We have actually tried other very advanced sensors that are more compact and thinner [1], but it takes a long time to make our customized gloves and they break very easily.
>
> **Why does this benchmark could be widely used.**
>
> - The tactile signals in our dataset contain basic features of tactile sensors, contact or not and the contact forces, which can facilitate many common daily tasks;
> - though there are different sensors in the community, they measure the contact force. If we convert their original signals to forces, different sensors can share the same methods. We provide the fitted function from voltage to forces in the supplementals;
> -  If the sensors are not converted to forces, the benchmark methods can be adapted to the target sensor type. For example, the patch tokens for tactile signals can be easily replaced with patches of tactile images from Gelsight, as in references 5 and 6 you mentioned.
> - We have tested that the visual-tactile model pretrained with the dataset of the single human hand can help not only single-hand manipulation tasks but also the bi-manual hand-over task via robotic hands, indicating the potential of tactile data and pretraining.
> - The tactile sensors we use are low-cost and easy to assemble into gloves. It is easy for the community to reproduce our work or collect more data.
> - Instead of pursuing high precision and high-resolution tactile data which is hard to achieve currently, we initiate the research of leveraging large-scale low-cost tactile data for pretraining for dexterous manipulation tasks. However, as mentioned in 3), the benchmark methods can be adapted with input encoding when better tactile signals can be acquired.
>
> I hope this helps clarify the contributions of our work.
>
> [1] Zhao, S., Li, Z., Xia, H., & Cui, R. (2023). Skin-Inspired Triple Tactile Sensors Integrated on Robotic Fingers for Bimanual Manipulation in Human-Cyber-Physical Systems. IEEE Transactions on Automation Science and Engineering.

---

> ### Author Response · Authors · 2024-11-22
> **Authors' response to Reviewer Z9Mf (2/3)**
>
> >There are some paper uses visual-tactile representations for reinforcement learning [5, 6]. I think compare using the pre-trained with those SOTA online-learning model is helpful (not only using simple ResNet and MLP).
>
> **Response to weakness 2:** Thank you for your insightful suggestion. Our work focuses on establishing a benchmark that incorporates models derived from existing research. The methods you mentioned are indeed valuable, and we will include them in future iterations of our benchmark.
>
> However, our benchmark primarily targets offline models. Models like CLIP have demonstrated promising trends in perception, and our experiments show that incorporating tactile signals (T) can yield good results. The challenge with online learning models is that they require retraining a representation model for each new task, which is time-intensive. In contrast, our benchmark is designed to leverage human-like vision-tactile demonstrations to pre-train a representation model that can generalize across diverse tasks, thus promoting task-level generalization.
>
> We will include a discussion of these methods in the related work to highlight their relevance and clarify our focus.
>
> >Section 5.4 shows that this framework can do Sim2Real and show some impressive demos. However, no quantitative results have been shown in the paper.
>
> **Response to weakness 3:** Thank you for raising these points regarding real-world experiments. Our paper proposes **a novel dataset and builds 17 benchmarking models upon existing works** for visual-tactile dexterous manipulation research. **Our primary focus is on benchmarking, not proposing a novel method. Therefore, we only provided some real-world demonstrations without presenting quantitative results.**
>
> To address your concerns, we conducted the real-world evaluations and the table below provides quantitative results for each task, with two objects per task. A trial is considered successful if the goal is achieved within 15 seconds. Trials exceeding the time limit or resulting in object drops are counted as failures. Each object is tested 10 times. We will include the quantitative results in the supplemental materials.
>
> | Tasks                         | BottleCap Turning | Faucet Screwing | Lever Sliding | Table Reorientation | In-hand Reorientation |
> | ----------------------------- | ----------------- | --------------- | ------------- | ------------------- | --------------------- |
> | Succ Times (20times in total) | 16                | 14              | 15            | 13                  | 10                    |
>
> >A new CoRL paper [7] present a method to learn tactile-guided control policy from human hand demonstrations by embedding tactile sensor on the human finger tip, where the idea is similar to collect human-tactile data with tactile glove shown in this paper. What's the difference between these two approaches? It's helpful if analyzing it in related work.
>
>
>
> **Response to weakness 4:** Thank you for pointing out this paper. At the time of our submission, this method had just been released, following the publication of the paper [1] we cited. Its emergence highlights the growing focus within the community on using human demonstrations to guide robotic manipulation. Compared to teleoperated data collection, this approach offers a more convenient way to obtain training data. Our dataset aligns well with this research trend, demonstrating the potential of human-tactile data for advancing dexterous manipulation. We appreciate your suggestion and will include a discussion of this work in related work.
>
> [1] Q. Liu, et al. "Masked Visual-Tactile Pre-training for Robot Manipulation," 2024 IEEE International Conference on Robotics and Automation (ICRA).

---

> ### Author Response · Authors · 2024-11-22
> **Authors' response to Reviewer Z9Mf (3/3)**
>
> >In section 5.2, the paper didn't show which task is it.
>
> **Response to weakness 5:** Thank you for your helpful comment. Due to the paper length limitation, we only show the results of the bottle cap task in Section 5.2. The results of other tasks are shown in the following table.
>
> | Method           | Faucet Screwing || Lever Sliding || Table Reorientation || In-hand Reorientation || Bimanual Hand-over || Task Mean |         |
> | ---------------- | --------------- | ------------- | ------------------- | --------------------- | ------------------ | --------- | -------- | -------- | --------- | -------- | -------- | -------- |
> |                  | Seen            | Unseen        | Seen                | Unseen                | Seen               | Unseen    | Seen     | Unseen   | Seen      | Unseen   | Seen     | Unseen   |
> | Base             | 49±12           | 43.9±10.5     | 5.8±4.4             | 2.2±1.9               | 36.9±12.8          | 27.6±10.5 | 38.1±2.4 | 33.7±1.6 | 8±4.4     | 3.3±1.4  | 32.3±1.6 | 24.6±1.7 |
> | T                | 56.9±15.8       | 62±14.3       | 44.2±18.6           | 40.3±16.7             | 57.9±6.7           | 57.3±7.2  | 48±1.5   | 37.7±1.2 | 25.1±10.5 | 14.8±6.4 | 50.8±2.5 | 47±2.1   |
> | V                | 60.2±16         | 56.3±16.3     | 0.4±0.3             | 0.2±0.2               | 20.2±16.3          | 18.8±15.3 | 60.5±0.6 | 55.8±1.2 | 2.5±0.9   | 2.2±1.1  | 24±3     | 22.2±2.9 |
> | V+T              | 60.3±15.3       | 54.2±14.8     | 15.1±13             | 0±0                   | 34.5±11            | 28±12.8   | 0.3±0.2  | 0±0      | 3.1±1.1   | 3.9±1.2  | 23.6±2.6 | 19.3±2.9 |
> | V-MVP            | 28.2±14.5       | 24.5±13.1     | 19.5±16.9           | 13.5±11.7             | 33.8±8.4           | 30±9.3    | 61.3±1.7 | 56.3±2.1 | 14.4±0.6  | 0±0      | 35.2±2.7 | 29.4±2.4 |
> | V-Voltron        | 13.8±6          | 12±5.7        | 79.7±6.8            | 65.2±7.3              | 45±15.6            | 35.4±12.8 | 60.1±4   | 48.8±4.9 | 0±0       | 0±0      | 40±1.9   | 31.7±1.5 |
> | V-R3M            | 67.3±7.2        | 58.5±6.7      | 20.4±7.4            | 15.3±4.1              | 48.1±11.6          | 36.2±10.2 | 7.6±6.5  | 12±10.4  | 21.7±6.8  | 0±0      | 37±0.7   | 26.2±2.1 |
> | V-CLIP           | 64.1±10.8       | 57±10.1       | 66.3±10.1           | 52.1±12               | 68.5±2.7           | 57.2±2.5  | 59.8±2.5 | 52.3±2.5 | 34±2.1    | 15±0.4   | 61.3±1.5 | 49.4±1.8 |
> | V-ResNet         | 63.2±4.2        | 64.1±4.5      | 74.7±4.2            | 61.2±4.1              | 71.7±3.5           | 65.4±2.3  | 2±0.7    | 2±0.7    | 35.3±1.9  | 17±1.2   | 54.1±0.5 | 46.8±0.6 |
> | V-MVP+T          | 25.1±14.5       | 22.7±13.1     | 4.8±4.1             | 12±6.7                | 35.2±17.3          | 37.9±17   | 60.4±1.7 | 54.1±1.4 | 29.7±2.3  | 6.4±3.4  | 38.5±2.5 | 35.3±2.3 |
> | V-Voltron+T      | 9.1±4.3         | 8±3.8         | 84.4±7.2            | 73.7±8                | 41.7±18.2          | 38±15.7   | 58±1.8   | 48.2±1.4 | 13.8±7.4  | 7.8±3.9  | 39.8±2.1 | 34.7±2   |
> | V-R3M+T          | 38.9±13.9       | 32.6±12.4     | 27.3±13.7           | 16.4±8.5              | 48.6±12.9          | 40.7±13.8 | 9.2±3.8  | 17±11.1  | 29.7±2.3  | 6.4±3.4  | 38.9±2.1 | 31±1.5   |
> | V-CLIP+T         | 73.7±3.7        | 66.8±4        | 74.2±12.9           | 64.5±13.4             | 75.2±1.1           | 67.4±1.6  | 55.6±2.2 | 48.3±1.8 | 35.6±9.1  | 17±3.8   | 65.4±1.7 | 55.9±1.7 |
> | V-ResNet+T       | 74.2±4.7        | 67.5±5        | 62.7±15.7           | 49.5±16.1             | 73.3±4.8           | 63.9±6.1  | 10.3±0.2 | 0±0      | 29.2±8    | 3.8±3.3  | 55.4±1.9 | 44.1±1.8 |
> | T-Pretrain       | 60±12.3         | 51.9±12.1     | 53.1±23.1           | 48.3±20.7             | 62.8±5.4           | 56.3±5.1  | 42.1±2.7 | 35.8±2.6 | 35±10.3   | 20.7±6   | 54.7±2.9 | 46.9±2.5 |
> | V-Pretrain       | 57.9±7          | 51.8±6.5      | 27.9±14.9           | 20.5±10.9             | 52.2±10.4          | 43.6±9.6  | 55.7±1.5 | 53.5±1.7 | 37.7±10.9 | 23.1±7   | 50.3±1.7 | 41.8±1.6 |
> | VT-JointPretrain | 80.1±1.8        | 73.6±2.1      | 89.3±3.6            | 79.6±6.1              | 84.4±2.2           | 78.2±1.9  | 62.5±5   | 55.1±2.7 | 45.5±1.5  | 26.6±1.9 | 74.3±0.6 | 65.7±0.7 |
>
>
> >This paper only compare the VT-JointPretrain with V-Pretrain, V-Pretrain+T, T-Pretrain, and V+T. How about using V-Pretrain+T-Pretrain separately?
>
> **Response to question 1:** Thank you for your suggestion. The table shows the results. We will add it to our benchmark.
>
> | Method                | Seen     | Unseen    |
> | --------------------- | -------- | --------- |
> | T-Pretrain            | 75.4±2.9 | 68.6±5.6  |
> | V-Pretrain            | 70.8±7.2 | 58.5±14.2 |
> | V-Pretrain+T-Pretrain | 78.5±3.7 | 74.8±5.7  |
> | VT-JointPretrain      | 83.7±0.9 | 81.3±0.5  |

---

> ### Comment · Reviewer_Z9Mf · 2024-11-23
>
> Thanks for the the response with more experimental results. The current version with more experimental results and justifications increase the quality of this paper. I will increase my score.
>
> I still have some questions and suggestions regarding to the response and the current version:
>
> 1. The explanation definitely highlight the contribution of this paper. However, although I understand that piezoresistive sensor is easier to be embedded into the human hand and human hand is the best embodiment for scaling up data collections, the question is that we still don't know whether this sensor is the best for dex hand or not. There are some other new hardware design for adding tactile sensors onto the dex hand: uskin[1] and Digit plexus[2]. Although they might be hard to be embedded onto the human's hand, but whether they are better for dexterous manipulation or not? Also, although we might not be able to scale up this kind of visual-tactile dataset with human hand, there already has some large pre-trained models for gelsight[3, 4] and can use MimicTouch[5] to scale up tactile data. It's also possible to combine a visual pre-trained model from human hand and those tactile pre-trained models. Then, which kind of sensor should we use? I think using visual-tactile pretraining with human hand dataset is a good direction, but I still has a bit concern that choosing to build a benchmark when none of this is clear may limit its contribution. Although this bench might be able to extend to different tactile sensors, it has not been proved yet.
>
> 2. Thanks for providing all the results in Sec. 5.2. My question is, in this draft, I think you didn't point out the task in this section is BottleCap
>
> [1]. https://www.xelarobotics.com/xela-integrations/allegro-hand-curved
>
> [2]. https://github.com/facebookresearch/digit-plexus
>
> [3]. Higuera et al., Sparsh: Self-supervised touch representations for vision-based tactile sensing, 2024
>
> [4]. Zhao et al., Transferable Tactile Transformers for Representation Learning Across Diverse Sensors and Tasks, 2024
>
> [5]. Yu et al., MimicTouch: Leveraging Multi-modal Human Tactile Demonstrations for Contact-rich Manipulation, 2024

---

> > ### Author Response · Authors · 2024-11-23
> >
> > **Response to question 1:**  Thank you for the comments, which inspire us to dig deeper into the tactile choice.
> >
> > 1. For the question of “Which kind of sensor should we use?”, we actually do not have an answer for it. Or, the answer is that different sensors will be used. Similar to robotic end-effectors including suction grippers, two-finger, three-finger, and multi-finger hands commonly used, tactile sensors are likely to span a spectrum—from low-cost, simple designs to highly precise, specialized ones. **Heterogeneity may become one of the major challenges for general manipulations.**
> >
> > Unlike RGB images, which have become a standardized sensor for vision, tactile sensing is still under development and inherently heterogeneous, with various sensor designs tailored to different applications. As what we have agreed with, sensors like Gelsight, piezoresistive arrays, and emerging technologies such as uSkin and DIGIT Plexus, each offer unique advantages, but they also involve trade-offs in resolution, form factor, cost, and scalability. Though optical-based sensors like Gelsight have gained increasing attention in recent years, piezoelectric sensors used in the paper are among the top tactile Market value **(**[**https://www.gminsights.com/industry-analysis/tactile-sensors-market/market-analysis**](https://www.gminsights.com/industry-analysis/tactile-sensors-market/market-analysis)**).** Another reason we use piezoelectric sensors is other very popular tactile sensors like capacitive sensors, resistive sensors or other e-skin sensors all can convert the force into voltage signals. From the perspective of the final voltage signal, if the function between the voltage values and forces values are provided, the difference between these sensors would be akin to the difference between different RGB cameras: the heterogeneity problem becomes smaller. However, we still believe that embodied intelligence inherently requires diversity in both sensors and actuators.
> >
> >
> >
> > 2) We agree with your comments on combining different pre-trained vision or tactile models. The concurrent pre-trained work mentioned [2,3,4] uses optical-based sensors and we use piezoelectric sensors. We think the community needs some different trials. Though we have also tried optical-based tactile sensors as an option using teleoperation to collect multi-fingered data with Allegro hand via a hand exoskeleton, it is too slow and so hard to finish a task. Using teleoperation like ALOHA (two parallel grippers) is very slow compared with human collection and using robotic hands with more fingers makes it harder to collect data. We then abandon the teleoperation solution.
> >
> > The combination, or the exploration of heterogeneous tactile pretraining is not only valuable but potentially necessary. Regardless of sensor type, the underlying principle of tactile sensing lies in force interaction during hand-object contact, which is a shared foundation across all tactile modalities. Recent methods such as HPT[1] and T3[2] propose network architectures and learning strategies to handle heterogeneous actuator and sensor modalities. While our current work establishes a foundation using piezoresistive sensors, we hope it inspires subsequent efforts to expand benchmarks with heterogeneous sensor data, such as using the method [5] to collect tactile image during human manipulation. We think it requires collective community efforts to collect data for various tasks and integrate heterogeneous tactile datasets and develop pretraining strategies to support general human like manipulation.
> >
> > [1] Wang et al., Scaling proprioceptive-visual learning with heterogeneous pre-trained transformers. NeurIPS 2024.
> >
> > [2]. Zhao et al., Transferable Tactile Transformers for Representation Learning Across Diverse Sensors and Tasks, CoRL, 2024
> >
> > [3]. Higuera et al., Sparsh: Self-supervised touch representations for vision-based tactile sensing, CoRL, CoRL, 2024
> >
> > [4]. Yu et al., MimicTouch: Leveraging Multi-modal Human Tactile Demonstrations for Contact-rich Manipulation, CoRL, 2024
> >
> >
> >
> > **Response to question 2:** Thank you for your reminder. We will ensure this clarification is included in the revised draft to avoid any confusion.

---

> > > ### Comment · Reviewer_Z9Mf · 2024-11-23
> > >
> > > Thanks for your detailed response. I don't have any other questions now and believe this paper is suitable for this conference. I would like to increase my score.

---

> > > > ### Author Response · Authors · 2024-11-24
> > > >
> > > > Thank you again for taking the time to review our work! Your detailed evaluation and valuable suggestions have been incredibly helpful in improving our paper. We sincerely appreciate your support and feedback.

---

### Official Review · Reviewer_bKve · 2024-11-01

**Soundness:** 2
**Presentation:** 3
**Contribution:** 2
**Rating:** 5
**Confidence:** 4

**Summary:**

This paper proposes a vision-tactile dataset and a novel benchmark for evaluating the vision-tactile pre-training methods. The experiments show that the tactile signals can provide guidance for the deployment of the tactile modality.

**Strengths:**

1. The experiments demonstrate both simulation and real-world settings.

**Weaknesses:**

- This MAE-based method is not novel. There is also another paper try to use the MAE to fuse the vision and tactile information: The power of the senses: Generalizable manipulation from vision and touch through masked multimodal learning. So the author at least should compare with it.

- The comparison is not fair. The pre-trained model-based methods only use an MLP to concatenate with tactile information. This operation cannot illustrate that these models cannot leverage the tactile information better.

- Could you give me some intuitions about why VT-Joint training can improve the robustness of viewpoint changing? I think the reason might be these tasks heavily reply on tactile rather than vision.

- Why Voltron performs so bad?

- Where is the real-world experiment results?

**Questions:**

Seen above in weakness.

---

> ### Author Response · Authors · 2024-11-22
> **Authors' response to Reviewer bKve (1/2)**
>
> We sincerely thank the reviewer for taking the time to evaluate our work and provide valuable feedback. **The primary contribution of our work lies in introducing a novel dataset and benchmark** designed to study how human vision-tactile data can guide robots in learning dexterous manipulation skills, **rather than proposing a new method to solve this problem**. The benchmark includes methods based on existing work, such as MAE-based approaches, contrastive learning techniques, and ResNet-based models. We have not positioned method innovation as a core contribution of our work.
>
> For the concerns and questions you raised, we have provided detailed responses in the hope of clarifying the value of our contribution and its potential impact on advancing research in this field. Thank you again for your thoughtful suggestions, which have been instrumental in refining our work. The responses are listed as follows:
>
> > This MAE-based method is not novel. There is also another paper that tries to use the MAE to fuse the vision and tactile information: The power of the senses: Generalizable manipulation from vision and touch through masked multimodal learning. So the author at least should compare with it.
>
> **Response to weakness 1:** Thank you for your question. **Our paper is not intended to propose a novel method for vision-tactile fusion and we do not claim the MAE-based method as our contribution. Rather, our contribution is a dataset and a benchmark that evaluates how visual and tactile signals can be effectively utilized in dexterous manipulation**. This MAE-based method (VT-JointPretrain) with other MAE-based methods are among the 17 pre-trained and non-pre-trained approaches in our benchmark, across a variety of tasks to provide a comprehensive evaluation.
>
> Regarding the MAE-based method mentioned in your comment, **our benchmark includes comparisons with similar MAE-based approaches, such as MVP and Voltron, and also non-MAE-based like CLIP, Resnet**. The specific paper you referenced was published at IROS 2024 in October after our work had already been submitted to ICLR. Nonetheless, we appreciate your suggestion and will consider including it in our benchmark in future iterations to ensure broader coverage and comparison.
>
> > The comparison is not fair. The pre-trained model-based methods only use an MLP to concatenate with tactile information. This operation cannot illustrate that these models cannot leverage the tactile information better.
>
> **Response to weakness 2:** Thank you for your valuable feedback. We sincerely apologize if our paper caused any misunderstanding. **To clarify first, our contribution is the dataset and the benchmark. We did not imply that the use of these pre-trained models is suboptimal. On the contrary, some of these pre-trained models are highly effective as shown in our results. Integrating them into our benchmark was aimed at enriching the diversity of the benchmarking evaluation methods.**
>
> Our goal of the paper is to provide a foundation for exploring how these models and tactile signals can be utilized together. MLP is effective in dealing with tactile signals: as shown in Table 3, the success rate achieves about 70%. Therefore, in our benchmark, we combine pretrained models for vision and MLP for tactile signals. Joint-Pretrain vision and tactile tokens based on Pretrained large models may be better alternatives as VT-JointPretrain implies and it requires efforts from the community to explore the possible structures and training strategies, which is also the aim of our work.
>
> We hope our benchmark can support future research in this area by offering a reference for effectively combining pre-trained models with tactile signals for robotic manipulation tasks. It serves as a stepping stone for the community to explore better approaches for leveraging pre-trained models and tactile information.
>
> > Could you give me some intuitions about why VT-Joint training can improve the robustness of viewpoint changing? I think the reason might be these tasks heavily rely on tactile rather than vision.
>
> **Response to weakness 3:** Thank you for your question. The pre-training process integrates both visual and tactile information, simulating how humans rely on multiple senses. Thus, even when the viewpoint is suboptimal or there is significant visual occlusion, the tactile data helps compensate for the limitations of the visual input. This fusion reduces the impact of poor visual signals, thereby enhancing the model’s robustness to changes in viewpoint.

---

> ### Author Response · Authors · 2024-11-22
> **Authors' response to Reviewer bKve (2/2)**
>
> > Why Voltron performs so bad?
>
> **Response to weakness 4:** Great Spot. **The main reason is the pooling method**. Similar to V_MVP and V-Pretrain, Voltron is also an MAE-based method. In previous experiments, we found that using a CLS token as the vision-tactile representation input for downstream tasks yielded more consistent results than Voltron’s multiheaded attention pooling. Although Voltron discusses the best pooling methods in their paper, in our setup, using the CLS token or applying mean pooling, as in MVP, proved to be more effective. We show the V-Pretrain performance of using CLS token and multiheaded attention pooling (MAP) in the table. We embed V-Pretrain into bottlecap policy learning with the CLS token or using Multi-head attention pooling (MAP), indicating MAP is not a suitable pooling method in our downstream tasks.
>
> | Method                       | Seen     | Unseen    |
> | ---------------------------- | -------- | --------- |
> | CLS token                    | 70.8±7.2 | 58.5±14.2 |
> | Multi-head attention pooling | 21.8±9.5 | 19.7±10.1 |
>
> > Where is the real-world experiment results?
>
> **Response to weakness 3:** Thank you for raising these points regarding real-world experiments. Our paper proposes **a novel dataset and builds 17 benchmarking models upon existing works** for visual-tactile dexterous manipulation research. **Our primary focus is on benchmarking, not proposing a novel method. Therefore, we only provided some real-world demonstrations without presenting quantitative results.**
>
> To address your concerns, we conducted the real-world evaluations and the table below provides quantitative results for each task, with two objects per task. A trial is considered successful if the goal is achieved within 15 seconds. Trials exceeding the time limit or resulting in object drops are counted as failures. Each object is tested 10 times. We will include the quantitative results in the supplemental materials.
>
> | Tasks                         | BottleCap Turning | Faucet Screwing | Lever Sliding | Table Reorientation | In-hand Reorientation |
> | ----------------------------- | ----------------- | --------------- | ------------- | ------------------- | --------------------- |
> | Succ Times (20times in total) | 16                | 14              | 15            | 13                  | 10                    |

---

> ### Comment · Reviewer_bKve · 2024-11-24
> **reply to authors**
>
> 1. The work I mentioned was published last year in arxiv.  This paper has updated some results recently. Therefore, "The specific paper you referenced was published at IROS 2024 in October after our work had already been submitted to ICLR. " this is not an excuse.
>
> 2. Almost in every question, you emphasize that your contribution is to introduce a novel dataset and benchmark. From my perspective, I admit that you propose a good dataset and conduct many experiments. So that's the reason why I scored 5. But you didn't explore how to leverage this dataset or dig out the potential of combining the simulation and real-world data, etc.
>
> 3. I think if you claim your real-world dataset does help the training performance. You have to verify the effectiveness of this method in real world.
>
> 4. I have to remind the author: For a dataset and benchmark paper, **it is not enough** to merely use the existing framework and compare it with existing methods to showcase the good performance of training with the dataset. There are two ways to improve this paper to reach the standard of ICLR:  A. as I said, **propose a proper algorithm and framework** to explore how to leverage this data better; B. **Conduct real-world experiments thoroughly** to demonstrate your policy can improve real-world performance. Simulation is not enough. C. **mix up simulation and real-world data**. D. illustrates this dataset is as important as Open-embodiment or Droid.

---

> > ### Author Response · Authors · 2024-11-24
> >
> > >The work I mentioned was published last year in arxiv. This paper has updated some results recently. Therefore, "The specific paper you referenced was published at IROS 2024 in October after our work had already been submitted to ICLR. " this is not an excuse.
> >
> > **Response to comment 1:** We are sorry that we don’t think it’s necessary to include non-peer-reviewed methods in our benchmark. Even though the referenced work was accepted at IROS 2024, according to the review policy, papers accepted (**not published**) within four months of the submission deadline do not need to be compared. You can verify this in the reviewer guidelines (https://iclr.cc/Conferences/2025/ReviewerGuide).
> >
> > **Q:** Are authors expected to cite and **compare with very recent work?** What about non-peer-reviewed (e.g., ArXiv) papers? (updated on 7 November 2022)
> >
> > **A:** We consider papers contemporaneous if they are published within the last four months. That means, since our full paper deadline is October 1, if a paper was published (i.e., at a peer-reviewed venue) on or after July 1, 2024, authors are not required to compare their own work to that paper.
> >
> >  >Almost in every question, you emphasize that your contribution is to introduce a novel dataset and benchmark. From my perspective, I admit that you propose a good dataset and conduct many experiments. So that's the reason why I scored 5. But you didn't explore how to leverage this dataset or dig out the potential of combining the simulation and real-world data, etc.
> >
> > **Response to comment 2:** We’re not sure where the misunderstanding lies. **In Section 4.2**, we clearly explained that VT-JointPretrain, V-Pretrain, and T-Pretrain are models trained **using our dataset**. These models serve as baselines we designed to utilize the dataset, and **in Section 5.3**, we provided a detailed analysis of the VT-JointPretrain method **trained on our dataset**.  Therefore, **we’re unclear why you believe we didn’t explore how to utilize the dataset**.
> >
> > Regarding your comment about exploring the combination of simulation and real-world data, this is exactly what our benchmark has achieved. The pretraining phase uses real-world data, while the simulation phase involves simulated data. All the pretraining models in our benchmark follow this approach. Although the pretraining uses real-world data, it does not interfere with the ability to learn from simulated data.
> >
> >  >I think if you claim your real-world dataset does help the training performance. You have to verify the effectiveness of this method in the real world.
> >
> > **Response to comment 3:** Thanks again for your suggestions. We have supplemented the **response to weakness 3** with quantitative results from real-world experiments, further validating the effectiveness of our method.
> >
> >  >I have to remind the author: For a dataset and benchmark paper, it is not enough to merely use the existing framework and compare it with existing methods to showcase the good performance of training with the dataset. There are two ways to improve this paper to reach the standard of ICLR: A. as I said, propose a proper algorithm and framework to explore how to leverage this data better; B. Conduct real-world experiments thoroughly to demonstrate your policy can improve real-world performance. Simulation is not enough. C. mix up simulation and real-world data. D. illustrates this dataset is as important as Open-embodiment or Droid.
> >
> > **Response to comment 4:** Thanks for your reminder.
> >
> >
> >
> > A. We did not directly use existing methods. The use of the CLS token for pooling features for RL training was something we discovered while replicating current methods, and it is not present in the original methods. We have discussed this point in **Response to weakness 4.** Additionally, we have developed several vision-tactile variant models with other pre-trained vision models. Except for the purely visual models, all other models (pretrained visual model + tactile) were constructed by us, not existing in the previous works.
> >
> > B. We have provided real-world experimental results that validate the effectiveness of our benchmark.
> >
> > C and D. **Such comparisons are unfair and unnecessary as we study different problems.** The hybrid data and the datasets (**Open X-embodiment or Droid)** you mentioned are primarily related to two parallel gripper-based tasks.  The datasets are mainly collected by the robotic grippers. Due to decades of research on two parallel grippers, the grippers have already established the basic skills like grasping and now the community focuses more on using the data collected by these basic skills to study long horizon tasks. However, for dexterous multi-fingered hands, establishing these basic skills is still in the nascent stages. Our work studies the learning of these basic manipulation skills with multi-fingered hands. We think the comparison with work on different problems is not necessary and not straightforward to compare in the sense of “fair”.

---

### Official Review · Reviewer_WkCE · 2024-11-02

**Soundness:** 3
**Presentation:** 3
**Contribution:** 3
**Rating:** 6
**Confidence:** 3

**Summary:**

This paper introduces a new visual-tactile dataset for training and benchmarking dexterous manipulation skills. It includes 10 tasks with 182 objects collected by human operators and provides a comprehensive benchmark of 6 challenging dexterous manipulation tasks. Additionally, the paper presents an extensive evaluation of over 17 methods on the benchmark, showcasing the efficacy of the dataset. Overall, this research is a valuable contribution, offering a well-structured dataset and benchmark for advancing visual-tactile manipulation.

**Strengths:**

- The paper introduces a highly diverse visual-tactile dataset for complex dexterous manipulation tasks, enhancing the realism and diversity of training data.

- Detailed analysis and evaluation of the proposed dataset underscore its robustness and potential impact on advancing dexterous manipulation.

- The proposed VT-JointPretrain method is promising, effectively integrating visual and tactile inputs for manipulation tasks.

- The extensive experimental analysis provides insights into the impact of pre-training on dexterous manipulation, with comparisons between different pre-trained and non-pre-trained methods on the benchmark.

**Weaknesses:**

- The tactile data in the proposed dataset lacks high-resolution force measurements, providing only binary force information per sensor. This limitation may impact tasks requiring precise force control, such as in-hand rotation or object flipping, where the exact contact force is critical.

- An ablation study of the visual-tactile representation learning framework is necessary to understand its efficacy further. Specifically:
(i) Examining performance variations with different masking rates of image and tactile data would clarify the framework’s robustness.
(ii) Analyzing performance with only image or only tactile data masked could highlight the independent contributions of each modality.

- Although the authors demonstrate successful deployment of the trained policy on real robots, the evaluation of its real-world performance is limited and lacks detailed analysis. A more rigorous evaluation of real robotic setups would strengthen the claims of real-world applicability.

**Questions:**

1- How well do the learned skills transfer to real-world dexterous manipulation scenarios? Have you evaluated the performance on a real robotic setup, and if not, are there plans for such an evaluation?

2- Could you elaborate on the rationale for using binary force information in the tactile data? Do you have plans to incorporate higher-resolution force data in future versions of the dataset to improve its applicability for tasks requiring fine-grained force control?

3- Have you considered conducting an ablation study with varying mask rates or with only image/tactile modalities masked? If so, could you share preliminary results or insights on how these variations impact the performance of the VT-JointPretrain framework?

4- Given the variety of tasks in the benchmark, how well does the proposed framework generalize across different types of manipulation tasks? Are there specific tasks or object types where it struggles, and if so, what might be the underlying reasons? (further discussion on failures could improve the paper)

---

> ### Author Response · Authors · 2024-11-22
> **Authors' response to Reviewer WkCE (1/2)**
>
> We sincerely thank the reviewer for the detailed and constructive feedback, as well as for recognizing the value and contributions of our work. Our primary objective is to establish a comprehensive dataset and benchmark for evaluating and comparing the effectiveness of various vision-tactile integration methods across diverse manipulation tasks. As such, our analysis focuses on performance comparisons among methods and between visual and tactile modalities. For the concerns you raised, we provide the following responses to address them in detail.
> >  The tactile data in the proposed dataset lacks high-resolution force measurements, providing only binary force information per sensor. This limitation may impact tasks requiring precise force control, such as in-hand rotation or object flipping, where the exact contact force is critical.
>
> **Response to weakness 1:** Thank you for raising this important point. While the tactile data used in our pretraining and benchmarking experiments is binarized, we would like to clarify that **our dataset also includes raw contact force information**. For further details, please refer to Section A.1 in the supplementary materials.
>
> We agree with your opinion that precise force measurements are crucial for tasks requiring fine-grained force control. **We have** **discussed this in** ***Tactile Modality*** and ***Tactile Pretraining*** **in Section 5.5 (Lines 489 and 501)**. However, high-resolution sensors are currently expensive, bulky and hard to be equipped into a multi-fingered hand (either robotic or human hand), which poses significant challenges for large-scale data collection. Although precise force control is not explicitly validated in our benchmark, **our experiments demonstrate that binary force signals can significantly aid finger coordination—a critical aspect shared by all tasks in our paper, as they require complex finger collaboration to complete**.
>
> Furthermore, **our dataset includes continuous force signals**. While these signals may not achieve the precision of systems like GelSight, **the sensors used in our work can measure forces ranging from 20g to 2kg,** **covering the majority of forces required for everyday manipulation tasks**, even delicate operations such as picking strawberries.
>
> Our work serves as an initial exploration in this field. We believe that as sensor hardware advances to become smaller and more precise, our research can provide valuable insights and a solid foundation for integrating these next-generation sensors into vision-tactile dexterous manipulation systems.
> > An ablation study of the visual-tactile representation learning framework is necessary to understand its efficacy further. Specifically: (i) Examining performance variations with different masking rates of image and tactile data would clarify the framework’s robustness. (ii) Analyzing performance with only image or only tactile data masked could highlight the independent contributions of each modality.
>
> **Response to weakness 2:** Thank you for your insightful suggestion. Our paper primarily focuses on establishing a comprehensive platform for evaluating and comparing the effectiveness of different vision-tactile integration methods across various manipulation tasks. Due to space limitations, our paper primarily focuses on modality analysis rather than an in-depth exploration of network structure. However, we have conducted the experiments you mentioned. For the visual modality, we used a masking ratio of 0.75 (used in MAE), while for the tactile modality, we experimented with three different masking ratios: 0.25, 0.5, and 0.75. Additionally, we have supplemented the experiments referenced in (ii), and the results of all experiments are summarized in the table below.
>
>
>
> | Method             | Seen     | Unseen   |
> | ------------------ | -------- | -------- |
> | tmr025_vmr075      | 81.2±0.8 | 80.5±1.2 |
> | **tmr05_vmr075\*** | 83.7±0.9 | 81.3±0.5 |
> | tmr075_vmr075      | 80±2.4   | 76±1.8   |
> | tmr0_vmr1          | 78.3±0.6 | 76.7±2.3 |
> | tmr1_vmr0          | 75±3.4   | 66.9±4.1 |
>
> （**tmr** and **vmr** are the mask ratio of tactiles and images. * is the parameter used in our benchmark）

---

> ### Author Response · Authors · 2024-11-22
> **Authors' response to Reviewer WkCE (2/2)**
>
> > Although the authors demonstrate successful deployment of the trained policy on real robots, the evaluation of its real-world performance is limited and lacks detailed analysis. A more rigorous evaluation of real robotic setups would strengthen the claims of real-world applicability.
>
> **Response to weakness 3:** Thank you for raising these points regarding real-world experiments. Our paper proposes **a novel dataset and builds 17 benchmarking models upon existing works** for visual-tactile dexterous manipulation research. **Our primary focus is on benchmarking, not proposing a novel method. Therefore, we only provided some real-world demonstrations without presenting quantitative results.**
>
> To address your concerns, we conducted the real-world evaluations and the table below provides quantitative results for each task, with two objects per task. A trial is considered successful if the goal is achieved within 15 seconds. Trials exceeding the time limit or resulting in object drops are counted as failures. Each object is tested 10 times. We will include the quantitative results in the supplemental materials.
>
> | Tasks                         | BottleCap Turning | Faucet Screwing | Lever Sliding | Table Reorientation | In-hand Reorientation |
> | ----------------------------- | ----------------- | --------------- | ------------- | ------------------- | --------------------- |
> | Succ Times (20times in total) | 16                | 14              | 15            | 13                  | 10                    |
> > How well do the learned skills transfer to real-world dexterous manipulation scenarios? Have you evaluated the performance on a real robotic setup, and if not, are there plans for such an evaluation?
>
> **Response to question 1:** Thank you for your question. The response aligns with our explanation in **Response to weakness** 3, which addresses this concern in detail. We hope this alleviates your concerns.
>
>
>
> > Could you elaborate on the rationale for using binary force information in the tactile data? Do you have plans to incorporate higher-resolution force data in future versions of the dataset to improve its applicability for tasks requiring fine-grained force control?
>
> **Response to question 2:** Thank you for your thoughtful question. As mentioned in [1], there are three different forms of finger coordination—simple synergies, reciprocal synergies, and sequential patterns. These coordination patterns lead to distinct interaction modes between the hand and the object, resulting in different contact patterns across various regions of the hand. The binarized signals we use represent whether there is contact between the fingers and the object. By analyzing these binary signals, we can describe the contact patterns associated with different coordination types of fingers. And we point out at line 64 that “though these high-precision tactile signals help sophisticated control at the low level, the combination and the tempo of touch status of different hand parts may provide abundant information on how to manipulate in a higher planning level”.
>
> Great suggestion **for the future plan**. We have developed a soft and high-resolution tactile sensor by embedding flexible pressure sensors into a soft silicone layer, allowing to be seamlessly adapted to multi-finger robotic hands and human hands for collecting tactile signals without compromising finger dexterity. We are utilizing this advanced sensor for data collection, aiming to enrich our dataset with more detailed and diverse tactile perception signals and enhancing its applicability for tasks requiring fine-grained force control.
>
> [1] Elliott, J. M., & Connolly, K. J. (1984). A CLASSIFICATION OF MANIPULATIVE HAND MOVEMENTS. Developmental Medicine & Child Neurology, 26(3), 283–296.
> > Have you considered conducting an ablation study with varying mask rates or with only image/tactile modalities masked? If so, could you share preliminary results or insights on how these variations impact the performance of the VT-JointPretrain framework?
>
> **Response to question 3:** Thank you for your question. The response is the same as **Response to weakness 2**.
>
> > Given the variety of tasks in the benchmark, how well does the proposed framework generalize across different types of manipulation tasks? Are there specific tasks or object types where it struggles, and if so, what might be the underlying reasons? (further discussion on failures could improve the paper)
>
> **Response to question 4:** Thank you for your question. We have discussed this issue in Section 5.5 of the paper, **Downstream Task Adaptation**, where we discuss the adaptation of VT-JointPretrain across various manipulation tasks. This section includes an analysis of task-specific challenges and potential limitations, highlighting cases where the framework may encounter difficulties.

---

### Official Review · Reviewer_xGyB · 2024-11-04

**Soundness:** 3
**Presentation:** 2
**Contribution:** 3
**Rating:** 5
**Confidence:** 4

**Summary:**

The author introduces a unique dataset for visual-tactile with 10 daily tasks.  A new benchmark with six complex dexterous manipulation tasks is proposed, using a reinforcement learning-based vision-tactile skill learning framework.
The authors have key findings that incorporating tactile data boosts task success rates by up to 40%.

**Strengths:**

1. I appreciate the authors' efforts to collect a larger visuo-tactile dataset.
2. Sufficient experiments as well as real experiments.
3. The visualization for t-SNE is good to show how tactile differ in different tasks.

**Weaknesses:**

Major Weaknesses:
1. The overall presentation and low readability.

2. The contribution 2 is questionable. Half of the tasks have been already widely used in previous benchmark tasks[1], like bottlecap turning, faucet screwing, reorientation, and bimanual hand-over. Even though previous work is state-based not using visual-tactile, the authors cannot mention their contribution to creating tasks.

3. The authors do real-world experiments, but there is no table of experiments to show the number, which cannot make any convincing conclusion.


[1] Chen, Y., Wu, T., Wang, S., Feng, X., Jiang, J., Lu, Z., ... & Yang, Y. (2022). Towards human-level bimanual dexterous manipulation with reinforcement learning. Advances in Neural Information Processing Systems, 35, 5150-5163.

**Questions:**

1. [line 382-387] Please make clear the definition of robustness to the viewpoint. Usually, when we refer to robustness to the viewpoint, we are talking about training policy in one viewpoint and test on another viewpoint.

2. Please specify the meaning and describe a little bit for each subfigure of Figure 1 and 2, instead of just putting on sentence in the caption.

3. [Line 16]. Only 6 tasks have been found in the paper, but the authors mention they have 10 tasks.

4. The real-world experiments use teacher-student to train the policy. However, this is not mentioned in the method section, which may cause confusion.

5. In section 5 benchmarking study, please specify how many seen objects are used and how many unseen objects are used. It would be better for the author to show all the objects that are used for each task.

---

> ### Author Response · Authors · 2024-11-22
> **Authors' response to Reviewer xGyB (1/2)**
>
> We sincerely thank the reviewer for their thoughtful feedback. As suggested, we have revised the paper and uploaded the updated version. We address these concerns and clarify potential misunderstandings. We have the following responses.
> > The overall presentation and low readability.
>
> **Response to weakness 1:** Thank you for your feedback regarding the overall presentation and readability. We will carefully revise the manuscript to improve its clarity and ensure that the paper is presented in a more accessible and reader-friendly manner. We appreciate your suggestion and will make the necessary changes in the revised version
> > The contribution 2 is questionable. Half of the tasks have been already widely used in previous benchmark tasks[1], like bottlecap turning, faucet screwing, reorientation, and bimanual hand-over. Even though previous work is state-based not using visual-tactile, the authors cannot mention their contribution to creating tasks.
>
> **Response to weakness 2:** Thank you for your comment. **We would like to clarify that our paper does not claim to create all tasks from scratch**. As stated in line 103, our primary contribution is providing a benchmark platform designed for vision-tactile dexterous manipulation research, including six complex dexterous tasks. And we said “there exists work on some similar tasks like Reorientation” in line 88-89.
>
> For tasks, **our bottlecap task involves turning the cap with a robotic hand, while [1] focuses on pulling it off** and there is no complex movement of turning/screwing.  Additionally, **the Faucet Screwing and Lever Sliding tasks in our benchmark are original** and not present in [1]. **The other three tasks are sourced from existing works**. We modified observation spaces and adjusted reward structures to better support vision-tactile modality research. All tasks emphasize generalization across objects, ensuring robustness during evaluation. [1] is a benchmark for bimanual dexterous manipulation, where many tasks exist in the previous works, such as hand-over you mentioned is sourced from [2].
>
> We hope this clarifies our contributions and addresses your concerns. If any wording caused confusion, we welcome your feedback and are open to revising it for clarity.
>
> [1] Chen, Y., Wu, T., Wang, S., Feng, X., Jiang, J., Lu, Z., ... & Yang, Y. Towards human-level bimanual dexterous manipulation with reinforcement learning. Advances in Neural Information Processing Systems, 2022.
>
> [2] Charlesworth, H. J., & Montana, G. (2021, July). Solving challenging dexterous manipulation tasks with trajectory optimisation and reinforcement learning. In International Conference on Machine Learning (pp. 1496-1506). PMLR.
> > The authors do real-world experiments, but there is no table of experiments to show the number, which cannot make any convincing conclusion.
>
> **Response to weakness 3:** Thank you for raising these points regarding real-world experiments. Our paper proposes **a novel dataset and builds 17 benchmarking models upon existing works** for visual-tactile dexterous manipulation research. **Our primary focus is on benchmarking, not proposing a novel method. Therefore, we only provided some real-world demonstrations without presenting quantitative results.**
>
> To address your concerns, we conducted the real-world evaluations and the table below provides quantitative results for each task, with two objects per task. A trial is considered successful if the goal is achieved within 15 seconds. Trials exceeding the time limit or resulting in object drops are counted as failures. Each object is tested 10 times. We will include the quantitative results in the supplemental materials.
>
> | Tasks                          | BottleCap Turning | Faucet Screwing | Lever Sliding | Table Reorientation | In-hand  Reorientation |
> | ------------------------------ | ----------------- | --------------- | ------------- | ------------------- | ---------------------- |
> | Succ Times  (20times in total) | 16                | 14              | 15            | 13                  | 10                     |
> > [line 382-387] Please make clear the definition of robustness to the viewpoint. Usually, when we refer to robustness to the viewpoint, we are talking about training policy in one viewpoint and test on another viewpoint.
>
> **Response to question 1:** Thank you for pointing this out. You are correct, "robustness to viewpoint" may be a bit misleading in this context. We will revise the manuscript to clarify that **we are referring to "viewpoint adaptability" rather than "robustness."** We appreciate your suggestion and will update the text accordingly.

---

> ### Author Response · Authors · 2024-11-22
> **Authors' response to Reviewer xGyB (2/2)**
>
> > Please specify the meaning and describe a little bit for each subfigure of Figure 1 and 2, instead of just putting on sentence in the caption.
>
> **Response to question 2:** Thank you for your suggestion. We will update the captions for Figures 1 and 2 to provide a brief explanation of each subfigure, ensuring that the reader can easily interpret the figures without needing to refer to the main text.
> > [Line 16]. Only 6 tasks have been found in the paper, but the authors mention they have 10 tasks.
>
> **Response to question 3:** Thank you for pointing that out. To clarify, the number mentioned in **line 6** refers to the number of tasks in the human manipulation dataset we collected. **Our dataset indeed contains 10 tasks in total, and the 6 tasks referenced in the paper correspond to the robot manipulation tasks** we validated in the downstream experiments.
> > The real-world experiments use teacher-student to train the policy. However, this is not mentioned in the method section, which may cause confusion.
>
> **Response to question 4:** Thank you for your comment. **Our work is a dataset and a benchmark, not a paper with novel methods. Therefore, the real-world experiment is not our focus**. The real-world experiments were conducted to demonstrate the deployment of task-specific policies trained on our benchmark in practical scenarios. **The details of the teacher-student training framework used in the real-world experiments are provided in Sec.C of Supplementary Material.**
>
> > In section 5 benchmarking study, please specify how many seen objects are used and how many unseen objects are used. It would be better for the author to show all the objects that are used for each task
>
> **Response to question 5:** Thank you for your comment. The details of the number of seen and unseen objects used in the benchmarking study, along with the specific objects for each task, are **provided in Section 4.1**. Additionally, **you can find visual examples of the object manipulation results in Figure 5 and 6**, which showcase the objects used in the experiments.

---

### Author Response · Authors · 2024-11-29
**Response Letter: Summary of Revisions and Updates**

**Dear Reviewers,**

Thank you for taking the time to review our work. Based on your valuable feedback, we have made several modifications and re-uploaded the updated PDF. The changes are summarized as follows:

- **For all reviewers:**

  - We added a table summarizing qualitative results in the real world (Appendix C.3).
  - Figure 2 has been updated to list all the baselines in the benchmark.
  - We added an anonymous URL in the abstract to allow anyone to reproduce our work.


- **For Reviewer xGyB:**

  - We revised "robustness to the viewpoint"      to "viewpoint adaptation."
  - Detailed explanations of the figure contents have been added to the legend.
  - To provide a clearer understanding of the experimental objects, we added an image of the items in Figure 6.

- **For Reviewer bKve:**

  - We added discussions of recent works in the related work section.
  - In the experimental discussion, we clarified that the comparison highlights the advantages of modality fusion over simple concatenation rather than comparing the overall performance of pretrained models.

- **For Reviewer Z9Mf:**

  - We added discussions of recent works in the related work section.
  - Table 3 has been updated, and we explicitly indicated which tasks the results correspond to within the paper.

We have addressed all the issues raised by the reviewers through detailed revisions and responses. If you have any further questions or concerns, we welcome additional discussion and will respond promptly and thoroughly.

Thank you again for your thoughtful feedback.

---

### Author Response · Authors · 2024-12-03
**General response**

We thank all the reviewers for their thoughtful feedback and valuable insights.

We are encouraged that they recognize our work as “**a valuable contribution**, offering a well-structured dataset and benchmark for advancing visual-tactile manipulation,” “**a highly diverse visual-tactile dataset** for complex dexterous manipulation tasks, enhancing the realism and diversity of training data,” and that “the dataset covers a wide variety of tasks and objects, which **makes it valuable for generalizing robotic manipulation skills**.” Additionally, we appreciate the acknowledgment that “the benchmark evaluates various pretraining and non-pretraining methods, **providing a comprehensive understanding of how different modalities affect task performance**.”

We are also pleased to note the positive reception of our experimental design: “**Sufficient experiments** as well as real experiments,” “**Detailed analysis and evaluation** of the proposed dataset underscore its robustness and potential impact on advancing dexterous manipulation,” and “The extensive experimental analysis **provides insights into the impact of pre-training** on dexterous manipulation.”

Following the reviewers’ constructive suggestions, we acknowledge that our work has room for improvement and have addressed all reviewers' concerns. In response:

1. **For clarity**, we have revised the caption used in figure legends and experiment names to avoid ambiguity.
2. **For experiments**, we have added an analysis of the mask ratio for the **VT-JointPretrain** method and introduced a new baseline, **V-Pretrain+T-Pretrain**, into our benchmark. We added more real-world experiments with quantitative results.
3. **For related work**, we have expanded the discussion to include recent advancements in the field.
4. **For reproducibility,** we added an anonymous URL in the abstract to allow anyone to reproduce our work.

This year, two studies on visual-tactile dexterous manipulation have been published in *Science Robotics*, underscoring the importance of visual-tactile integration in robotic skill learning. Our work **contributes a new perspective to this domain by leveraging human visual-tactile information to guide downstream robot learning for dexterous manipulation.** We present **a diverse human manipulation visual-tactile dataset**, covering everyday tasks with first-person visual perspectives and tactile force information. Building on existing efforts, we construct **a benchmark that includes 17 baselines and supports six complex dexterous manipulation tasks**. Unlike prior works that focus on a single task, **our benchmark emphasizes diversity in data, models, and tasks**, providing a richer foundation for advancing visual-tactile manipulation research.

We believe the diversity of our dataset and benchmark can inspire new ideas and directions for visual-tactile dexterous manipulation. We hope our work can facilitate and spur further research in this important field. Thank you again for your valuable time and feedback.

Kind regards,

 *All the authors*

---

### Meta-Review · Area_Chair_TPed · 2024-12-26

**Metareview:**

The paper introduces a vision-tactile dataset for complex robotic manipulation skill learning, along with a benchmark of six dexterous manipulation tasks and a reinforcement learning framework. The study demonstrates that integrating visual and binary tactile modalities improves task success rates, adaptability to unknown tasks, and robustness to noise and deployment settings.

The reviewers expressed mixed opinions about the paper. They acknowledged its contributions, including (1) the collection of the vision-tactile dataset and the benchmark design, (2) the intuitive integration of visual and tactile inputs, (3) a thorough evaluation in both simulation and real-world settings, and (4) the demonstrated performance boost from joint pretraining of vision and touch. However, they also raised concerns regarding (1) the novelty of the method, (2) the quality of the paper’s presentation, (3) questions about the claimed contributions, and (4) the need for more detailed analysis and quantitative evaluation of performance in both simulation and real-world experiments.

During the Author-Reviewer Discussion phase, the authors effectively addressed many concerns by updating tables and figures, providing additional experiments and discussions, and clarifying key points, which convinced some reviewers to raise their scores. After thoroughly reviewing the reviews, the rebuttal, and the discussions, the AC recognizes the paper’s value, particularly from a dataset and benchmark perspective, and believes it represents a meaningful addition to the robot learning community. However, concerns remain regarding the presentation quality, the potential for additional real-world experiments, and missing comparisons with additional baselines.

The AC recommends acceptance of the paper but urges the authors to carefully address the reviewers’ comments in a revised version. Specifically, the authors should consider conducting additional hardware experiments, as limited access to similar hardware may hinder reproducibility and broader adoption. For dataset and benchmark contributions, ensuring proper maintenance and widespread adoption is often more impactful than the initial publication itself.

**Additional Comments On Reviewer Discussion:**

The Reviewer Discussion phase involved extensive debate. Reviewer bKve raised concerns about a lack of novelty, insufficient real-world experiments, and missing comparisons with recent work. Reviewer Z9Mf provided a detailed rebuttal, arguing that while these concerns are valid, they do not warrant rejection, as the paper still offers significant value to the conference. The AC concurs with this assessment and encourages the authors to refine the paper and follow up to maintain the benchmark/dataset to maximize its impact.

---

### Decision · Program_Chairs · 2025-01-22

Accept (Poster)